# Zero Sum SVD: Balancing Loss Sensitivity for Low Rank LLM Compression

Ali Abbasi [1]   Chayne Thrash [* 1]   Haoran Qin [* 1]   Shansita Sharma [1]   Sepehr Seifi [1]   Soheil Kolouri [1 2 3]

## Abstract

Advances in large language models have driven strong performance across many tasks, but their memory and compute costs still hinder deployment. SVD-based compression reduces storage and can speed up inference via low-rank factors, yet performance depends on how rank is allocated under a global compression ratio. Prior methods often use homogeneous ranks for similarly sized matrices, despite large differences in loss sensitivity, or rely on expensive iterative pre-truncation optimization to determine per matrix ranks. We propose **Zero Sum SVD** (**ZS-SVD**), a post-training method that performs *global* singular component selection using activation whitening and first-order calibration loss estimates in whitened coordinates. **ZS-SVD** prunes components across the whole model with a **zero sum** rule that keeps the cumulative predicted loss change near zero, automatically yielding heterogeneous ranks without solving a rank allocation optimization. Motivated by evidence that gradients near pretrained solutions exhibit low rank structure, we also introduce an optional lightweight correction that applies a **single** projected gradient update after truncation, followed by re-truncation. Extensive experiments across multiple LLM architectures show consistent gains across diverse benchmarks and compression ratios. Code is available at https://github.com/mint-vu/Zero-Sum-SVD

## 1. Introduction

Large Language Models (LLMs) have demonstrated remarkable capabilities across natural language understanding and generation tasks (et al., 2025a; Meta AI, 2025; et al., 2025b). Despite their success, the deployment of LLMs remains constrained by substantial computational and memory requirements, limiting their adoption on resource-constrained devices and latency-sensitive applications such as robotics, edge computing, and interactive systems (Wan et al., 2024; Zhu et al., 2024). This motivates the design of post-training compression methods that reduce model size without expensive retraining.

Quantization and knowledge distillation are often effective at reducing model cost while preserving performance. However, quantization methods (Frantar et al., 2023; Lin et al., 2025; Dettmers et al., 2023; Liu et al., 2025) frequently depend on specific hardware features or kernel support to realize their speedups, which can limit portability across deployment environments. Knowledge distillation (Ko et al., 2025; Hsieh et al., 2023; Gu et al., 2024; Fang et al., 2025), while powerful, typically requires training a separate student model via gradient-based optimization, adding substantial computational and engineering overhead compared to purely post-training alternatives. Structured pruning approaches (Ma et al., 2023; Ashkboos et al., 2024; Frantar & Alistarh, 2023; Guo et al., 2025) can also suffer meaningful accuracy drops even under moderate pruning ratios.

In contrast, Singular Value Decomposition (SVD) offers a compelling alternative that circumvents these limitations. SVD-based compression is *hardware-agnostic*, requiring only standard linear algebra operations; *flexible*, enabling compression to arbitrary target ratios; and compatible with other compression approaches, enabling combination with quantization and distillation. Moreover, SVD-compressed models can reduce runtime KV cache memory without additional accuracy loss (Yuan et al., 2025).

Early SVD-based compression methods either (i) apply standard low-rank factorization to reconstruct the weight matrix itself (Jaderberg et al., 2014; Ben Noach & Goldberg, 2020), or (ii) use importance-weighted reconstructions—e.g., Fisher-weighted low-rank factorization as in FWSVD (Hsu et al., 2022). In both cases, however, the objective is still local: it prioritizes matching $W$ (under a chosen metric) rather than directly optimizing task loss. This matrix-centric view can miss a key reality in LLMs: performance is shaped by the *distribution of inputs* each layer

---

[*]Equal contribution  [1]Department of Computer Science, College of Connected Computing, Vanderbilt University, Nashville, TN. [2]Department of Electrical and Computer Engineering, Vanderbilt University, Nashville, TN. [3]Nanofold.ai. Correspondence to: Ali Abbasi <ali.abbasi@vanderbilt.edu>, Soheil Kolouri <soheil.kolouri@vanderbilt.edu, soheil.kolouri@nanofold.ai>.

*Proceedings of the 43rd International Conference on Machine Learning*, Seoul, South Korea. PMLR 306, 2026. Copyright 2026 by the author(s).

actually sees. As a result, truncating directions that appear negligible under a matrix norm can still cause large, systematic shifts in typical pre-activations $WX$ on in-distribution tokens, which can compound across layers and worsen under higher compression rates.

Recent variations of SVD-based compression (Yuan et al., 2025; Wang et al., 2025b; Qinsi et al., 2025; Ding et al., 2025), incorporate activation statistics by whitening and performing truncation in the corresponding coordinate system, which better aligns the approximation with the empirical distribution of $x$ from a small calibration set. However, the scope of their truncation objective is still largely local, focusing on reconstructing $Wx$ or closely related surrogates, rather than directly characterizing how truncation changes the end-to-end loss, including error propagation through subsequent layers and interactions across modules. DipSVD (Ding et al., 2025) takes a step toward loss awareness by introducing a dual-importance protection term based on per-matrix Fisher information. While promising, the resulting importance factor is heuristic and only indirectly related to the true loss change, making its impact harder to predict across models, tasks, and compression regimes.

We address these limitations with **Zero-Sum SVD (ZS-SVD)**, a post-training compression method that makes truncation decisions using both the local distortion induced in the whitened activation space and a global, loss-aware view derived from first-order calibration gradients. Our main contributions are: 1) We introduce a *global* zero-sum selection strategy that prunes singular components across all layers while keeping the cumulative predicted loss drift near zero, yielding heterogeneous ranks automatically under a global budget without solving an explicit rank-allocation optimization. 2) Motivated by evidence that gradients near pretrained solutions exhibit low-rank structure, we propose an optional lightweight correction step that briefly deviates from the low-rank manifold via a projected gradient update on a small calibration set, and then re-truncates to return to the target rank while incurring minimal additional projection error. We validate ZS-SVD through extensive experiments on LLMs spanning multiple scales (e.g., 7B, 13B, 30B) across perplexity and zero-shot reasoning benchmarks, demonstrating consistent improvements over strong SVD baselines (SVD-LLM, Dobi-SVD) and structured pruning methods, along with meaningful inference speedups relative to prior SVD approaches.

## 2. Related Work

**Compression of Large Language Models.** LLM performance has scaled with model size, but at the cost of heavy memory and compute that complicate deployment. This has driven post-training compression methods that avoid full retraining, most commonly pruning and quantization.

Optimal Brain Compression (OBC) (Frantar et al., 2022) performs greedy post-hoc pruning/quantization by minimizing activation reconstruction error on a small calibration set, but is impractical for LLMs because it requires repeated inverse-Hessian computations. SparseGPT (Frantar & Alistarh, 2023) scales this idea to 100B+ models, while LLM-Pruner (Ma et al., 2023) uses neuron connectivity for structured pruning and Wanda (Sun et al., 2024) proposes a one-shot magnitude–activation criterion. On the quantization side, LLM.int8() (Dettmers et al., 2022) uses mixed precision to handle activation outliers; SmoothQuant (Xiao et al., 2023) mitigates outliers by rescaling weights using activation statistics; GPTQ/OPTQ (Frantar et al., 2023) adapts OBC with fixed row-wise ordering and block Hessian approximations; and QuIP (Chee et al., 2023; Tseng et al., 2024) further improves accuracy via incoherence processing. Despite strong compression, many pruning and quantization methods yield limited wall-clock speedups without specialized hardware and kernel support.

**SVD-based Compression of Large Language Models** Recently, Singular Value Decomposition (SVD) based compression approaches have gained popularity due to their simplicity and ability to realize inference speed improvements regardless of hardware. FWSVD (Hsu et al., 2022) initially applied SVD-based compression to LLMs. To reduce the error incurred by this truncation, parameter importance is computed via the Fisher information matrix. Neurons are then weighted by the sum of their weights' importance scores prior to computing the SVD. Since then, works have primarily focused on removing singular values such that activation reconstruction error is reduced. ASVD (Yuan et al., 2025) rescales each weight matrix using a diagonal matrix of per-channel average activation magnitudes before SVD truncation. SVD-LLM (Wang et al., 2025b) directly optimizes activation reconstruction by utilizing a "whitening" matrix computed as the Cholesky decomposition of the covariance of the input activations. In addition, they introduce a performance recovery step by fine-tuning the resulting low-rank matrices via a small LoRA (Hu et al., 2022; Wang et al., 2024) residual. SVD-LLMv2 (Wang et al., 2025a) extends this to support heterogeneous rank allocation per layer via utilizing an estimate of the loss incurred by truncation. Most recently, Dobi-SVD (Qinsi et al., 2025) uses backpropagation to optimize a layer-wise truncation level (rank) and computes the corresponding truncated approximation using incremental PCA (IPCA).

Our method is also SVD-based and activation-aware (in the spirit of SVD-LLM (Wang et al., 2025b;a)), but differs in how it globally allocates rank: we compute per-layer singular-value importances via the directional derivative of the calibration loss with respect to each singular value, inducing a cross-layer priority ordering that enables fast greedy, heterogeneous truncation without solving the ex-

pensive per-layer optimization used in methods such as Dobi-SVD (Qinsi et al., 2025). Finally, we augment truncation with a lightweight one-step correction followed by re-truncation, recovering loss while preserving the target low-rank structure.

## 3. Preliminaries

### 3.1. Notation and compression objective

Let $W \in \mathbb{R}^{m \times n}$ denote a weight matrix in a neural network, e.g., in a transformer block, and let $X \in \mathbb{R}^{n \times T}$ denote the corresponding input activation matrix formed by stacking $T$ $n$-dimensional tokens. We consider post-training compression that replaces $W$ by $W'$ while minimizing the activation reconstruction error

$$\min_{W'} \|WX - W'X\|_F, \tag{1}$$

where $\| \cdot \|_F$ denotes the Frobenius norm. We write the singular value decomposition (SVD) of a matrix $A \in \mathbb{R}^{m \times n}$ as $A = U\Sigma V^\top$, where $U \in \mathbb{R}^{m \times m}$ and $V \in \mathbb{R}^{n \times n}$ are orthogonal matrices whose columns are the left and right singular vectors of $A$, and $\Sigma \in \mathbb{R}^{m \times n}$ is a (rectangular) diagonal matrix whose diagonal entries $\{\sigma_i\}_{i=1}^r$ are the singular values of $A$, ordered as $\sigma_1 \geq \sigma_2 \geq \cdots \geq 0$, and where $r = \min(m, n)$; for a target rank $k \leq r$, the rank-$k$ truncated SVD is given by $A_k = U_k \Sigma_k V_k^\top$, where $U_k \in \mathbb{R}^{m \times k}$ and $V_k \in \mathbb{R}^{n \times k}$ contain the top $k$ left and right singular vectors, respectively, and $\Sigma_k = \mathrm{diag}(\sigma_1, \ldots, \sigma_k) \in \mathbb{R}^{k \times k}$.

### 3.2. Whitening for activation reconstruction

Define the activation second moment $C = XX^\top \in \mathbb{R}^{n \times n}$. Expanding (1) gives a weighted Frobenius norm,

$$\|WX - W'X\|_F^2 = \mathrm{tr}\big((W - W')\, C\, (W - W')^\top\big)$$
$$= \|(W - W')\, S\|_F^2, \tag{2}$$

where $S \in \mathbb{R}^{n \times n}$ is any factor such that $SS^\top = C$, for instance, the Cholesky decomposition of $C$. Hence, for a fixed rank budget $k$, minimizing the activation reconstruction error is equivalent to finding a rank $k$ approximation of the whitened matrix $WS$ under the Frobenius norm.

### 3.3. Truncation aware whitening and whitened SVD

In practice $C$ is unknown and must be estimated from a calibration set. We therefore compute a numerically stable whitening factor $S \in \mathbb{R}^{n \times n}$ by taking the Cholesky factorization of $C + \lambda I$, where $\lambda > 0$ is a small ridge term added for numerical stability. Then, we define the whitened weight matrix

$$A = WS, \tag{3}$$

and compute its SVD, $A = U\Sigma V^\top$. For a target rank $k$, let $A_k = U_k \Sigma_k V_k^\top$ denote the truncated SVD decomposition.

Mapping back yields the compressed weight

$$W_k' = A_k S^{-1} = U_k \Sigma_k V_k^\top S^{-1}. \tag{4}$$

For implementation convenience, we parameterize the rank-$k$ approximation as $W_k' = W_u' W_v'$, with:

$$W_u' = U_k \Sigma_k^{1/2} \in \mathbb{R}^{m \times k}, \qquad W_v' = \Sigma_k^{1/2} V_k^\top S^{-1} \in \mathbb{R}^{k \times n}. \tag{5}$$

**Theorem 3.1** (Whitened truncation yields singular value reconstruction loss). *Assume $SS^\top = XX^\top$ and let $A = WS = U\Sigma V^\top$. Let $A_k$ be the rank-k truncation of $A$ and $W_k' = A_k S^{-1}$. Then*

$$\|WX - W_k'X\|_F^2 = \sum_{i > k} \sigma_i^2. \tag{6}$$

**Corollary 3.2** (Optimality of truncated SVD in the whitened space). *Under the assumptions of Theorem 3.1, $W_k'$ minimizes $\|WX - W'X\|_F$ over all rank k matrices $W'$. Equivalently, truncating the smallest singular values of $A = WS$ is optimal for activation reconstruction at rank k.*

Proofs of Theorem 3.1 and Corollary 3.2 are provided in Appendix A.

We argue that while keeping $\|WX - W'X\|_F$ small is desirable, this reconstruction error does not directly capture how rank truncation impacts the calibration loss. We therefore propose to quantify the contribution of the singular values of each whitened weight matrix $A$ to the loss, and to use this singular-value "importance" to guide rank truncation, while remaining mindful of computational cost.

## 4. Method

### 4.1. Gradient-based singular value sensitivity

We aim to assign an *importance* score to each singular value of the whitened matrix $A = WS$, so as to directly quantify its contribution to the calibration loss and to determine which components to prune under a global rank budget. Let $A = U\Sigma V^\top$ with singular values $\{\sigma_i\}_{i=1}^r$. Consider removing the $i$'th component, i.e., $\sigma_i \leftarrow 0$, while keeping the singular vectors fixed. This induces a perturbation

$$\Delta A_i = -\sigma_i\, u_i v_i^\top. \tag{7}$$

Our goal is to estimate the induced change in the calibration loss $\mathcal{L}$ via a first-order approximation around the current model, taken with respect to the singular values of $A$.

Let $G_W = \nabla_W \mathcal{L}$ denote the gradient with respect to the original weight $W$. Because $A = WS$, a perturbation $\Delta A$ corresponds to $\Delta W = \Delta A\, S^{-1}$, and the first order loss change satisfies,

$$\Delta \mathcal{L} \approx \langle G_W, \Delta W \rangle = \langle G_W, \Delta A\, S^{-1} \rangle = \langle G_W S^{-\top}, \Delta A \rangle. \tag{8}$$

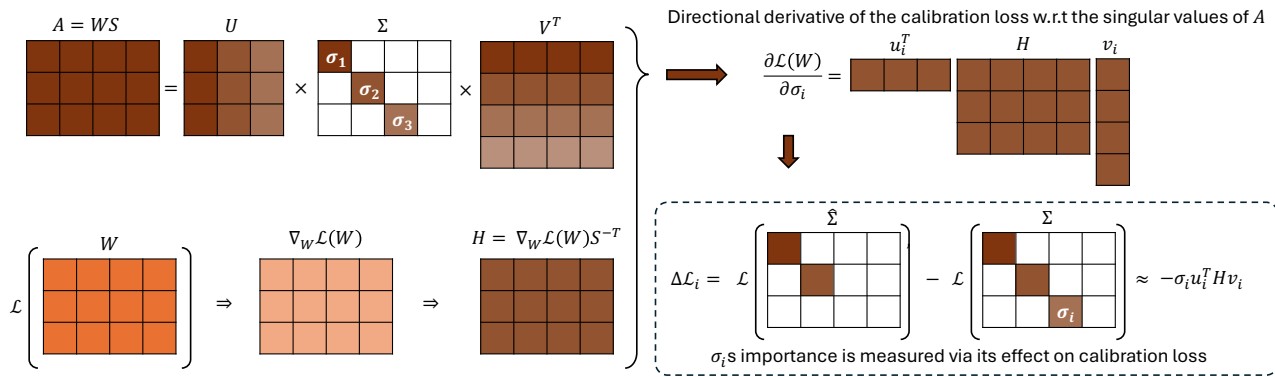

*Figure 1.* Zero-Sum SVD scores singular values of the whitened weight matrix using the calibration-loss directional derivative, then uses these scores to prioritize truncation across layers.

We then define $H = G_W S^{-\top}$ to be the whitened gradient, which can be efficiently computed when $S$ is triangular (e.g., Cholesky of $XX^T$). Substituting (7) into (8) yields an estimate for the loss change from dropping $\sigma_i$:

$$\Delta\mathcal{L}_i \approx -\sigma_i \langle H, u_i v_i^\top \rangle = -\sigma_i u_i^\top H v_i. \quad (9)$$

Collecting the per-component terms gives the vector

$$g_\sigma = \mathrm{diag}(U^\top HV) \in \mathbb{R}^r, \quad (10)$$

where $g_{\sigma,i} = u_i^\top H v_i$. In other words, $g_{\sigma,i}$ measures the first-order sensitivity of $\mathcal{L}$ to the singular value $\sigma_i$ in the whitened parameterization. Under the perturbation $\sigma_i \leftarrow 0$, the linearized loss change is $\Delta\mathcal{L}_i \approx -\sigma_i g_{\sigma,i}$, so both magnitude and sign matter: $|\sigma_i g_{\sigma,i}|$ quantifies the predicted impact, while $\mathrm{sign}(g_{\sigma,i})$ determines its direction. In particular, if $g_{\sigma,i} > 0$ then $\Delta\mathcal{L}_i < 0$, meaning that dropping component $i$ is predicted to *decrease* the calibration loss; if $g_{\sigma,i} < 0$, the loss is predicted to increase. This observation motivates the selection rule developed in the next section.

**4.2. Global budgeted truncation with zero sum selection**
Low-rank truncation compresses a dense $W \in \mathbb{R}^{m \times n}$ by storing $k(m + n)$ parameters instead of $mn$. Prior methods either fix per-layer ranks via a closed-form rule (e.g., $k = \lfloor \rho mn/(m + n) \rfloor$) (Wang et al., 2025b) or allocate ranks by solving a global optimization (Qinsi et al., 2025). We instead select individual singular components globally (in the whitened space), which naturally induces heterogeneous ranks across layers while matching a target retention rate, without expensive cross-layer optimization. We next introduce our zero-sum selection rule.

**Zero-sum selection rule.** We propose a greedy multi-layer truncation scheme that preserves per-layer spectral order while using signed first-order singular-value importance (in the whitened space) to keep the cumulative predicted loss change near zero. Within each matrix, we prune singular components from smallest to largest $\sigma_i$, so the next candidate is always the smallest remaining singular value. We

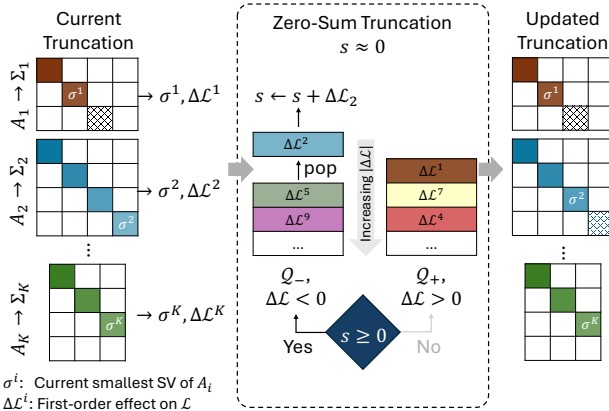

*Figure 2.* Zero-sum selection rule across different weight matrices.

maintain a global pool containing this next candidate from every matrix and iteratively select components so that the running sum of predicted loss changes stays close to zero.

Specifically, let $\mathcal{D}$ denote the set of components removed so far, and define the running cumulative predicted loss change as $s = \sum_{j \in \mathcal{D}} \Delta\mathcal{L}_j$, where $\Delta\mathcal{L}_i$ is defined in Eq. (9). At each iteration, we choose the next component so that its signed contribution tends to counteract the current drift of $s$ and move it back toward zero.

To implement this efficiently, we maintain two min-heaps partitioned by the sign of the predicted loss change: $\mathcal{Q}_+$ contains candidates with $\Delta\mathcal{L}_i \geq 0$ and $\mathcal{Q}_-$ contains candidates with $\Delta\mathcal{L}_i < 0$. Within each heap, candidates are keyed by $|\Delta\mathcal{L}_i|$, so among candidates of the chosen sign, we select the one with the smallest predicted impact.

Given the current running sum $s$, we choose a preferred sign and pop from the corresponding heap, falling back to the other heap if the preferred heap is empty:

$$\text{prefer } \mathcal{Q}_+ \text{ if } s \leq 0, \quad \text{prefer } \mathcal{Q}_- \text{ if } s > 0. \quad (11)$$

After selecting candidate $i$, we update $s \leftarrow s + \Delta\mathcal{L}_i$, remove the corresponding singular component from the correspond-

ing matrix, and insert that matrix's next available singular component into the appropriate heap.

This design couples a local ordering based on $\sigma_i$, which controls the reconstruction distortion in the whitened space, with a global ordering based on $\Delta\mathcal{L}_i$, which controls the predicted change in task loss. The combination strikes a balance between preserving activation level fidelity and preserving end task performance under a shared global budget.

**Stopping criterion.** Selection proceeds until the global parameter-removal budget is met. We maintain a running count of removed parameters: removing one singular component from an $m \times n$ weight corresponds to reducing the factor rank by one, saving $(m + n)$ parameters in the low-rank representation. Upon termination, the remaining components in each module determine its final rank, and we form the compressed weight by truncating $\Sigma$ accordingly and storing the resulting factors as in Eq. (5). We summarize the full procedure in Algorithms 1 and 2 (Appendix B).

### 4.3. Light correction step
Our compression pipeline alternates between *truncation* and, optionally, *correction*. We first truncate each weight matrix by projecting it onto a low-rank form to meet the global retention target, as described in the previous subsection. We may then apply a correction step that temporarily allows the weights to deviate from the low-rank manifold to recover expressivity and reduce task loss, after which we re-truncate (re-project) back to the target low-rank form. The key trade-off is that correction should meaningfully improve loss, yet remain sufficiently small so that the subsequent re-truncation does not incur a large projection error.

**One-Step Correction.** Let $W \in \mathbb{R}^{m \times n}$ be the original weight matrix and let $W'_k$ be its rank-$k$ truncation. Define the truncation residual $\Delta W \triangleq W - W'_k$, so that $W = W'_k + \Delta W$. Let $\mathcal{L}$ denote the calibration loss, and use the Frobenius inner product $\langle A, B \rangle \triangleq \text{tr}(A^\top B)$. A first-order expansion around $W'_k$ gives

$$\mathcal{L}(W) = \mathcal{L}(W'_k + \Delta W) \approx \mathcal{L}(W'_k) + \langle g, \Delta W \rangle, \quad (12)$$

where $g \triangleq \nabla_W \mathcal{L}(W'_k)$. We seek the smallest perturbation $\Delta$ to $W'_k$ that matches the same first-order loss change, i.e., $\langle g, \Delta \rangle = \langle g, \Delta W \rangle$. The minimum-Frobenius-norm solution is the projection of $\Delta W$ onto $g$:

$$\Delta W' \triangleq \frac{\langle g, \Delta W \rangle}{\langle g, g \rangle} g. \quad (13)$$

By construction, $\langle g, \Delta W' \rangle = \langle g, \Delta W \rangle$, hence from (12), $\mathcal{L}(W'_k + \Delta W) \approx \mathcal{L}(W'_k + \Delta W')$ up to first-order terms. We empirically compare this one-step correction against alternative correction strategies in Appendix B.1.

**Re-truncation.** After the one-step correction, the updated weight matrix is $W'_k + \Delta W'$. The following lemma upper-bounds the rank after an additive update.

**Lemma 4.1** (Rank bound under additive correction). *For any matrices $A$ and $B$ of compatible dimensions,* $\text{rank}(A + B) \leq \text{rank}(A) + \text{rank}(B)$.

By Lemma 4.1, if $\text{rank}(W'_k) = k$ and the correction satisfies $\text{rank}(\Delta W') \leq \ell$, then $\text{rank}(W'_k + \Delta W') \leq k + \ell$. Hence, after the one-step correction, we must re-truncate the corrected weights to enforce rank $k$. Importantly, when the correction $\Delta W'$ is low-rank (small $\ell$), the updated weights remain close to the rank-$k$ manifold, so the subsequent rank-$k$ projection incurs a smaller re-truncation error.

Interestingly, the rank of $\nabla_W \mathcal{L}(W) \in \mathbb{R}^{m \times n}$ has attracted recent attention, with growing evidence that gradients are often low-rank. This stems from the outer-product structure of backpropagation: for a fully connected layer, $\nabla_W \mathcal{L} = \sum_{i=1}^{B} \delta_i a_i^\top$, where $\delta_i$ is the backpropagated error and $a_i$ is the input for sample $i$ in a batch of size $B$ (Baker et al., 2024; Gelberg et al., 2025). Thus $\text{rank}(\nabla_W \mathcal{L}) \leq B$, and empirically the effective rank is often much smaller due to correlations in activations and error signals (Vogels et al., 2019). This has motivated methods that exploit gradient low-rankness, including PowerSGD (Vogels et al., 2019), LoRA (Hu et al., 2022), and GaLore (Zhao et al., 2024); see Balzano et al. (2025) for a survey.

We leverage this low-rank structure directly: since $\text{rank}(\Delta W') = \text{rank}(\nabla_W \mathcal{L}(W'_k)) \leq \ell$, and as established above, $\ell$ is typically very small in practice, re-truncation after our one-step correction introduces only negligible error.

### 4.4. Remapping
Note that the parameter ratio for a rank-$k$ SVD factorization, $\rho = \frac{k(m+n)}{mn}$, measures storage rather than rank. In particular, $\rho = 1$ implies $k = \frac{mn}{m+n} < \min(m, n) = \text{rank}(W)$ (e.g., for $m = n$, $k = n/2$), so the ratio saturates before full rank and is not in one-to-one correspondence with $k$ over $\rho \in (0, 1]$. Following Dobi-SVD (Qinsi et al., 2025), we also report *remap* variants that parameterize compression by the fraction of retained singular components. Specifically, for $\tilde{\rho} \in (0, 1]$ we set $k = \lfloor \tilde{\rho} \, \text{rank}(W) \rfloor \approx \lfloor \tilde{\rho} \min(m, n) \rfloor$. Dobi-SVD then uses a packed storage format in which the truncated factors $U_k \in \mathbb{R}^{m \times k}$ and $V_k \in \mathbb{R}^{n \times k}$ are implicitly encoded by storing only a modified $U_k$: assuming $m \geq n$, an 8-bit copy of $V_k$ is packed into the first $n$ rows of $U_k$, so the footprint scales as $k \cdot \max(m, n)$ (i.e., $mk$). Under this remapping, $\tilde{\rho}$ and $k$ are in one-to-one correspondence; see (Qinsi et al., 2025) for details.

**Remapping-aware truncation.** Remapping integrates directly into our global selection framework by modifying only the budget accounting. Under Dobi-SVD remapping, we store truncated factors $U_k \in \mathbb{R}^{m \times k}$ and $V_k \in \mathbb{R}^{n \times k}$ using their packing scheme. Assuming $m \geq n$, each rank-1 component contributes $(m - n)$ fp16 entries from the remaining rows of $U_k$, and $n$ fp8 entries that pack the first $n$

*Table 1.* ZS-SVD vs. SVD-based compression methods on LLaMA-7B across maintenance ratios (0.8/0.6/0.4). Lower is better for PPL; best is in bold. "1x/5x/10x" denote the number of truncate–correct–re-truncate iterations. (*) indicates results with Dobi-SVD-style remapping enabled. (†) denotes HQ (Half-prune+Quant), which we use in place of remapping at pruning ≥ 50%.

| RATIO | METHOD | PPL ↓ | | | ACC ↑ | | | | | | | AVG. ↑ | DROP ↓ |
|---|---|---|---|---|---|---|---|---|---|---|---|---|---|
| | | WIKI2 | PTB | C4 | OPENB. | ARC_E | ARC_C | WINOG. | HELLAS. | PIQA | MATHQA | | |
| 1.0 | BASELINE | 5.68 | 8.35 | 7.34 | 0.34 | 0.75 | 0.42 | 0.70 | 0.57 | 0.79 | 0.28 | 0.55 | 0.0% |
| 0.8 | ASVD | 11.14 | 16.55 | 15.93 | 0.25 | 0.53 | 0.27 | 0.64 | 0.41 | 0.68 | 0.24 | 0.43 | 21.8 |
| | SVD-LLM | 7.94 | 16.22 | 15.84 | 0.22 | 0.58 | 0.29 | 0.63 | 0.43 | 0.69 | 0.24 | 0.44 | 20.0 |
| | DOBI-SVD | 8.54 | 14.83 | 10.01 | 0.26 | 0.59 | 0.31 | 0.66 | 0.44 | 0.70 | 0.23 | 0.46 | 16.4 |
| | ZS-SVD | 6.74 | 11.87 | 10.74 | 0.31 | 0.70 | 0.37 | 0.68 | 0.49 | 0.74 | 0.24 | 0.50 | 9.1 |
| | ZS-SVD 1X | 6.61 | 11.26 | 10.57 | 0.31 | 0.72 | 0.39 | 0.67 | 0.49 | 0.74 | 0.25 | 0.51 | 7.3 |
| | ZS-SVD 5X | 6.43 | 11.17 | 10.42 | 0.32 | 0.72 | 0.39 | 0.67 | 0.50 | 0.75 | 0.25 | 0.51 | 7.3 |
| | DOBI-SVD* | 6.08 | 15.39 | 7.83 | 0.27 | 0.65 | 0.37 | 0.68 | 0.54 | 0.77 | 0.27 | 0.51 | 7.3 |
| | ZS-SVD* | 5.90 | 8.81 | 7.95 | 0.35 | 0.74 | 0.41 | 0.70 | 0.56 | 0.78 | 0.26 | **0.54** | **1.8** |
| 0.6 | ASVD | 1407 | 3292 | 1109 | 0.13 | 0.28 | 0.22 | 0.48 | 0.26 | 0.55 | 0.19 | 0.30 | 45.5 |
| | SVD-LLM | 13.11 | 63.75 | 49.83 | 0.19 | 0.42 | 0.25 | 0.58 | 0.33 | 0.60 | 0.21 | 0.37 | 32.7 |
| | DOBI-SVD | 13.54 | 46.38 | 23.54 | 0.22 | 0.41 | 0.27 | 0.58 | 0.34 | 0.61 | 0.23 | 0.38 | 30.9 |
| | ZS-SVD | 11.44 | 43.19 | 34.13 | 0.23 | 0.52 | 0.26 | 0.62 | 0.35 | 0.64 | 0.22 | 0.41 | 25.5 |
| | ZS-SVD 1X | 9.96 | 37.13 | 27.76 | 0.24 | 0.56 | 0.28 | 0.61 | 0.37 | 0.64 | 0.22 | 0.42 | 23.6 |
| | ZS-SVD 5X | 9.45 | 36.52 | 26.20 | 0.23 | 0.56 | 0.29 | 0.62 | 0.38 | 0.66 | 0.22 | 0.42 | 23.6 |
| | DOBI-SVD* | 8.12 | 43.85 | 12.63 | 0.28 | 0.65 | 0.32 | 0.62 | 0.45 | 0.72 | 0.25 | 0.47 | 14.5 |
| | ZS-SVD* | 6.96 | 12.72 | 11.52 | 0.32 | 0.71 | 0.36 | 0.68 | 0.48 | 0.74 | 0.24 | **0.50** | **9.1** |
| 0.4 | ASVD | 57057 | 45218 | 43036 | 0.12 | 0.26 | 0.21 | 0.49 | 0.26 | 0.53 | 0.18 | 0.29 | 47.3 |
| | SVD-LLM | 53.74 | 438.58 | 345.49 | 0.14 | 0.28 | 0.22 | 0.50 | 0.27 | 0.55 | 0.21 | 0.31 | 43.6 |
| | DOBI-SVD | 46.18 | 238.91 | 190.62 | 0.15 | 0.31 | 0.20 | 0.52 | 0.28 | 0.54 | 0.22 | 0.32 | 41.8 |
| | ZS-SVD | 45.17 | 334.85 | 212.57 | 0.15 | 0.31 | 0.21 | 0.54 | 0.28 | 0.55 | 0.21 | 0.32 | 41.8 |
| | ZS-SVD 1X | 26.92 | 222.93 | 121.32 | 0.17 | 0.35 | 0.21 | 0.54 | 0.29 | 0.56 | 0.21 | 0.33 | 40.0 |
| | ZS-SVD 5X | 20.41 | 163.58 | 88.56 | 0.17 | 0.37 | 0.22 | 0.55 | 0.30 | 0.57 | 0.21 | 0.34 | 38.2 |
| | ZS-SVD 10X | 18.49 | 144.89 | 76.86 | 0.17 | 0.38 | 0.23 | 0.55 | 0.30 | 0.58 | 0.22 | 0.35 | 36.4 |
| | DOBI-SVD* | 9.95 | 67.62 | 17.94 | 0.23 | 0.52 | 0.24 | 0.56 | 0.38 | 0.65 | 0.23 | 0.40 | 27.3 |
| | ZS-SVD† | 6.73 | 11.78 | 10.69 | 0.29 | 0.72 | 0.38 | 0.68 | 0.49 | 0.75 | 0.25 | **0.51** | **7.3** |

rows of $U_k$ together with the corresponding entries of $V_k$ ($n$ additional fp8). Hence, dropping one singular component reduces storage by $2(m-n) + 2n = 2m$ bytes, which corresponds to $\max(m,n)$ fp16-equivalent parameters. We therefore set $\mathrm{cost}(\ell) = \max(m_\ell, n_\ell)$ for remap-aware selection, while leaving the rest of the algorithm unchanged.

## 5. Experiments

**Models, datasets, and evaluation.** We evaluate along two axes: language modeling quality (perplexity) and downstream reasoning (zero-shot accuracy). Perplexity is reported on WikiText2 (Merity et al., 2017), Penn Treebank (Marcus et al., 1993), and C4 (Raffel et al., 2020); zero-shot accuracy on OpenBookQA (Mihaylov et al., 2018), ARC-Easy/Challenge (Clark et al., 2018), WinoGrande (Sakaguchi et al., 2021), HellaSwag (Zellers et al., 2019), PIQA (Bisk et al., 2020), and MathQA (Amini et al., 2019), computed with LM-Evaluation-Harness (Gao et al., 2024). Lower perplexity and higher accuracy are better; we also report the mean accuracy over the seven tasks and its relative drop from the uncompressed model. Following prior work, we truncate only the main transformer linear matrices (attention projections (q,k,v,o) and MLP layers) and use a

*Table 2.* PPL on WikiText2 (W2), PTB, and C4, plus Avg on 7 commonsense tasks, at 30% pruning.

| METHOD | LLAMA-7B | | | | VICUNA-7B | | | |
|---|---|---|---|---|---|---|---|---|
| | W2 | PTB | C4 | AVG | W2 | PTB | C4 | AVG |
| ASVD | 95.3 | 200.9 | 86.3 | 0.36 | 91.4 | 415.6 | 136.2 | 0.32 |
| FWSVD | 33.0 | 53.6 | 38.2 | 0.39 | 43.7 | 239.3 | 64.8 | 0.36 |
| SVD-LLM | 9.5 | 29.0 | 26.4 | 0.40 | 12.4 | 124.5 | 39.5 | 0.40 |
| DIP-SVD | 9.4 | 22.3 | 19.9 | 0.44 | 12.1 | 81.1 | 28.8 | 0.43 |
| ZS-SVD | 8.2 | 19.6 | 16.8 | 0.46 | 10.2 | 48.0 | 21.8 | 0.46 |

WikiText2 calibration set of 256 sequences of length 2048, matching SVD-LLM (Wang et al., 2025b) and Dobi-SVD.

**Perplexity and accuracy comparison.** Table 1 reports our main results on LLaMA-7B under maintenance ratios 0.8, 0.6, and 0.4, corresponding to truncating 20%, 40%, and 60% of the target weights, respectively. We compare against strong SVD-based baselines, including ASVD (Yuan et al., 2025), SVD-LLM (Wang et al., 2025b), and Dobi-SVD (Qinsi et al., 2025). Across all ratios, our zero-sum component selection consistently outperforms these baselines, improving both perplexity and zero-shot accuracy.

We also evaluate our optional correction procedure, which alternates truncation with a calibration-driven update while

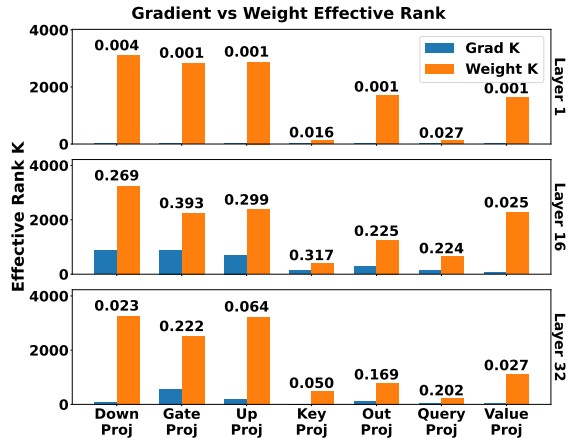

*Figure 3.* The numbers above each bar indicate the effective-rank ratio $k_{0.95}(G)/k_{0.95}(W')$ (grad over weight).

preserving low rank. After truncating to rank-$k$, we apply a one-step correction and then re-truncate to remove any rank growth; we repeat this truncate–correct–re-truncate cycle for 1, 5, or 10 updates. More updates consistently improve perplexity and mean accuracy, with the largest gains under aggressive compression, especially at maintenance ratio $0.4$.

Because Dobi-SVD reports most results with remapping enabled (except for LLaMA-7B), we adopt the same setting when comparing against Dobi-SVD. To enable fair comparison with methods that do not apply remapping, we also report non-remapped results for our method and all baselines. Since remapping effectively combines truncation with quantization, for pruning ratios $\geq 50\%$ we consider two footprint-matching strategies: (i) Dobi-SVD remapping, or (ii) prune to half the target ratio and uniformly halve the bit-width of all target parameters, which yields the same overall footprint reduction. The latter performs better under aggressive compression, and we denote it HQ (Half-prune + Quantize). For pruning ratios $< 50\%$, we follow the standard Dobi-SVD remapping protocol. More generally, both remapping and HQ are practical mechanisms for coupling truncation and quantization under a footprint constraint; improved joint budget allocation remains an interesting direction for future work.

Table 2 compares our method against Dip-SVD (Ding et al., 2025), FWSVD (Hsu et al., 2022), SVD-LLM (Wang et al., 2025b), and ASVD (Yuan et al., 2025) under $30\%$ pruning on LLaMA-7B and Vicuna-7B. We report perplexity on WikiText-2, PTB, and C4, together with the average accuracy over our commonsense reasoning suite. We include Dip-SVD as a recent gradient-informed baseline, but it relies on a heuristic per-matrix importance score rather than estimating first-order loss changes for individual singular components. Because Dip-SVD had no official implementation available at the time of writing, we run ZS-SVD under their stated $30\%$ pruning protocol and report our results alongside their reported numbers.

*Table 3.* ZS-SVD vs. pruning methods on LLaMA-2-7b.

| RATIO | METHOD | ACC ↑ | | | | | AVG ↑ |
|---|---|---|---|---|---|---|---|
| | | PIQA | HELL | WIN | ARC_E | ARC_C | |
| 1.0 | BASELINE | 0.78 | 0.57 | 0.69 | 0.76 | 0.43 | 0.65 |
| 0.6 | LLM-PRUNER | 0.70 | 0.41 | 0.53 | 0.53 | 0.27 | 0.48 |
| | SLICEGPT | 0.65 | 0.57 | 0.60 | 0.43 | 0.32 | 0.51 |
| | BONSAI | 0.72 | 0.45 | 0.58 | 0.59 | 0.30 | 0.53 |
| | WANDA-SP | 0.70 | 0.42 | 0.53 | 0.57 | 0.29 | 0.50 |
| | SVD-LLM | 0.56 | 0.30 | 0.57 | 0.39 | 0.21 | 0.41 |
| | ZS-SVD | 0.63 | 0.34 | 0.60 | 0.46 | 0.25 | 0.45 |
| | DOBI-SVD* | 0.72 | 0.45 | 0.64 | 0.67 | 0.31 | 0.56 |
| | ZS-SVD* | 0.72 | 0.46 | 0.67 | 0.66 | 0.33 | 0.57 |
| 0.4 | SVD-LLM | 0.54 | 0.27 | 0.48 | 0.26 | 0.20 | 0.35 |
| | ZS-SVD | 0.54 | 0.27 | 0.52 | 0.29 | 0.19 | 0.36 |
| | DOBI-SVD* | 0.67 | 0.38 | 0.57 | 0.55 | 0.26 | 0.49 |
| | ZS-SVD† | 0.73 | 0.48 | 0.68 | 0.70 | 0.36 | 0.59 |

*Table 4.* ZS-SVD vs. pruning methods on LLaMA-13b.

| RATIO | METHOD | ACC ↑ | | | | | AVG ↑ |
|---|---|---|---|---|---|---|---|
| | | BOOLQ | PIQA | WIN | ARC_E | ARC_C | |
| 1.0 | BASELINE | 0.78 | 0.79 | 0.73 | 0.77 | 0.47 | 0.70 |
| 0.8 | LLM-PRUNER | 0.67 | 0.77 | 0.65 | 0.68 | 0.38 | 0.63 |
| | FLAP | 0.70 | 0.78 | 0.69 | 0.73 | 0.43 | 0.66 |
| | SVD-LLM | 0.63 | 0.73 | 0.69 | 0.69 | 0.35 | 0.62 |
| | ZS-SVD | 0.77 | 0.77 | 0.70 | 0.73 | 0.43 | 0.68 |
| | DOBI-SVD* | 0.69 | 0.79 | 0.72 | 0.76 | 0.47 | 0.68 |
| | ZS-SVD* | 0.77 | 0.79 | 0.73 | 0.75 | 0.46 | 0.70 |

Tables 3 and 4 compare our method against popular structured pruning baselines, including LLM-Pruner (Ma et al., 2023), SliceGPT (Ashkboos et al., 2024), Bonsai (Dery et al., 2024), Wanda-sp (Sun et al., 2023), and FLAP (An et al., 2024), on LLaMA-2-7B and LLaMA-13B, respectively. Across pruning ratios and models, ZS-SVD consistently attains higher accuracy in commonsense reasoning.

Table 5 evaluates ZS-SVD under $20\%$ compression across OPT-6.7B, Vicuna-7B, and LLaMA-30B, with LLaMA-30B included to test scalability. We report WikiText-2 perplexity and average commonsense accuracy. Across all three architectures, ZS-SVD achieves a better perplexity–accuracy trade-off than prior SVD baselines, indicating that our global selection strategy generalizes across model families/sizes.

**Ablation: Does correction stay low-rank?** To answer this question and motivated by evidence that gradients near pretrained solutions are low-rank, we compare the spectra of the truncated weights and their gradients at the truncated point. We first truncate the model to a given pruning ratio, yielding $\{W'_\ell\}$. We then compute the calibration loss $\mathcal{L}$ on a mini-batch of 4 sequences (length 2048) from the same calibration set and run a single backward pass to obtain per-module gradients averaged over the $4 \times 2048$ tokens, $G_\ell = \nabla_{W_\ell} \mathcal{L}(W'_\ell)$. For each target matrix, we compute the

*Table 5.* Perplexity (PPL, ↓) on WikiText-2 and average accuracy (Acc, ↑) on six commonsense reasoning datasets (excluding arc_c) for OPT-6.7B, Vicuna-7B, and LLaMA-30B at 20% pruning.

| | OPT-6.7B | | Vicuna-7B | | LLaMA-30B | |
|---|---|---|---|---|---|---|
| Method | PPL | Acc | PPL | Acc | PPL | Acc |
| Original | 10.86 | 0.52 | 6.78 | 0.56 | 4.10 | 0.61 |
| SVD | 66275 | 0.03 | 18644 | 0.05 | 946.31 | 0.33 |
| FWSVD | 14559 | 0.06 | 2758 | 0.09 | 15.98 | 0.42 |
| ASVD | 82.00 | 0.32 | 16.23 | 0.33 | 6.74 | 0.44 |
| SVDLLM | 16.04 | 0.41 | 8.41 | 0.51 | 6.61 | 0.54 |
| ZS-SVD | 11.40 | 0.51 | 8.08 | 0.53 | 4.83 | 0.59 |

*Table 6.* Ablation of global $\sigma$ selection strategies on LLaMA-7B. We report WikiText-2 PPL at 0.4 and 0.6 retention rates.

| Strategy | Per-$W$ $\sigma$ sorted | Ratio 0.4 | Ratio 0.6 |
|---|---|---|---|
| Most negative $\Delta\mathcal{L}$ | × | 160594 | 373585 |
| Magnitude of $\Delta\mathcal{L}$ | × | 341.3 | 88.7 |
| Most negative $\Delta\mathcal{L}$ | ✓ | 182452 | 369350 |
| Magnitude of $\Delta\mathcal{L}$ | ✓ | 51.8 | 12.0 |
| Magnitude of $\sigma$ | ✓ | 803599 | 32750 |
| Zero-sum $\Delta\mathcal{L}$ (ZS-SVD) | ✓ | 45.2 | 11.4 |

*Table 7.* Throughput and memory for LLaMA-2-7B under different compression ratios on Slow GPU and Regular GPU.

| Comp. | Method | Titan Xp | | | RTX A5000 | | |
|---|---|---|---|---|---|---|---|
| | | Thro.-put (tok/s) | Peak Mem. (GB) | Act Mem. (GB) | Thro.-put (tok/s) | Peak Mem. (GB) | Act Mem. (GB) |
| 0% | Original | 15.57 | 2.15 | 0.02 | 130.03 | 13.56 | 1.00 |
| 40% | SVDLLM | 34.54 | 8.89 | 1.02 | 675.24 | 15.86 | 8.00 |
| | DobiSVD | 32.63 | 10.07 | 1.02 | 692.49 | 17.07 | 8.02 |
| | ZS-SVD | 35.21 | 8.81 | 1.02 | 731.68 | 15.78 | 8.00 |
| 60% | SVDLLM | 41.25 | 6.48 | 1.02 | 752.95 | 13.49 | 8.00 |
| | DobiSVD | 36.30 | 8.27 | 1.02 | 722.35 | 15.26 | 8.02 |
| | ZS-SVD | 40.66 | 6.39 | 1.02 | 762.34 | 13.37 | 8.00 |

singular values of $W'_\ell$ and $G_\ell$ and define the effective rank at energy threshold $\tau = 0.95$ as

$$k_\tau(A) = \min\left\{ k : \frac{\sum_{i=1}^{k} \sigma_i(A)^2}{\sum_j \sigma_j(A)^2} \geq \tau \right\}, \quad (14)$$

where $\{\sigma_i(A)\}$ are sorted in decreasing order. We report the effective-rank of weights and gradients and their ratio, $k_\tau(G_\ell)/k_\tau(W'_\ell)$, for Layers 1, 16, and 32 of LLaMA-2-7B at 20% pruning in Fig. 3. The gradients exhibit a small effective rank, supporting our claim that the correction step can be re-truncated with small projection error.

**Ablation: Inference efficiency on GPUs.** We benchmark LLaMA-2-7B inference on an RTX A5000 (25 GB) and a TITAN Xp (12 GB), comparing ZS-SVD against SVD-LLM, DobiSVD, and the uncompressed baseline (Table 7). On the RTX A5000, ZS-SVD delivers the highest throughput among compressed methods and provides large speedups over the uncompressed baseline (5.63× at 40% compression and 5.86× at 60%), while maintaining similar peak and activation memory to competing SVD approaches. For the

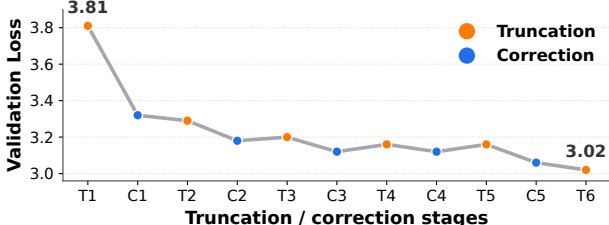

*Figure 4.* Validation loss (Wiki-2) across alternating truncation ($T_i$) and correction ($C_i$) stages, on LLaMA-7B at 60% pruning.

*Table 8.* Coupling truncation with quantization at 60% pruning on LLaMA-7B (WikiText-2 PPL, ↓).

| Method | Truncation | Remapping | HQ |
|---|---|---|---|
| SVD-LLM | 53.74 | 14.98 | 7.95 |
| Dobi-SVD | 46.18 | 9.95 | 8.79 |
| ZS-SVD | **45.17** | **9.25** | **6.73** |

uncompressed baseline in the low-VRAM regime, we reduce the sequence length to fit the full model, which lowers the reported activation and peak memory. On the memory-limited TITAN Xp, where the baseline requires CPU–GPU weight offloading, ZS-SVD improves throughput by 2.26× (40%) and 2.61× (60%) relative to the offloaded baseline, and attains lower peak memory than DobiSVD. Overall, these results show that ZS-SVD remains effective across both compute-friendly and memory-constrained regimes.

**Ablation: Different $\sigma$ pruning strategies.** Table 6 compares global selection rules for removing singular components under a fixed budget. "Per-$W$ $\sigma$ sorted" indicates whether each matrix enforces spectral order (the next candidate is its smallest remaining $\sigma$) or allows arbitrary removal. We evaluate: (i) *Most negative* $\Delta\mathcal{L}$, which greedily drives the cumulative predicted loss change negative; (ii) *Smallest* $|\Delta\mathcal{L}|$; (iii) *Smallest* $\sigma$ (ignoring loss); and (iv) *Zero-sum* $\Delta\mathcal{L}$ (ZS-SVD), which alternates positive/negative $\Delta\mathcal{L}$ to keep the running sum near zero while respecting per-$W$ ordering. ZS-SVD performs best at both ratios, suggesting that combining per-matrix spectral order with signed loss sensitivity is critical. Results are on LLaMA-7B with WikiText-2 perplexity. We see that the proposed zero-sum strategy outperforms others with a large margin.

**Ablation: Remapping vs. HQ.** Both remapping and HQ (Half-prune + Quantize) couple SVD truncation with 8-bit quantization to meet a target pruning ratio. HQ retains 2× the parameters before truncation and quantizes them to 8-bit, so the target compression is reached through truncation plus quantization. As 8-bit quantization alone already provides 50% compression, the only reasonable regime for HQ is at pruning ratios above 50%. Table 8 compares ZS-SVD, SVD-LLM, and Dobi-SVD at 60% pruning on LLaMA-7B (WikiText-2) under three schemes: truncation only, vanilla Dobi-SVD remapping, and HQ. HQ consistently outperforms remapping across all methods, and ZS-SVD is best

*Table 9.* Truncation time analysis on LLaMA-7B, with PPL evaluated on WikiText2.

| | SVDLLM | DOBISVD | ZS-SVD |
|---|---|---|---|
| **TIME** ($\downarrow$) | 7.9 MIN | 19.25 HR | 15.9 MIN |
| **PPL** ($\downarrow$) | 53.74 | 46.18 | 45.17 |

*Table 10.* First- vs. second-order sensitivity on LLaMA-7B (WikiText-2 PPL, $\downarrow$). Diagonal Fisher is a lightweight second-order proxy.

| PRUNING | 20% | 40% | 60% |
|---|---|---|---|
| FIRST-ORDER (OURS) | **6.74** | **11.44** | **45.17** |
| DIAGONAL FISHER | 6.83 | 11.98 | 46.32 |

under all three schemes.

**Ablation: First- vs. second-order Hessian.** Our importance score is first-order; the full Hessian is $P \times P$ in the parameter count $P$ and infeasible for LLMs, so we test the diagonal Fisher information as a lightweight second-order proxy. Table 10 compares the two on LLaMA-7B at 20%, 40%, and 60% pruning (WikiText-2): the first-order criterion is consistently better, likely because diagonal Fisher ignores off-diagonal curvature. Richer approximations, such as Kronecker-factored curvature, could be promising directions for future work.

**Truncation time analysis.** Table 9 reports end-to-end truncation time and WikiText-2 perplexity for SVD-LLM, Dobi-SVD, and ZS-SVD under the same setting. ZS-SVD is significantly faster than Dobi-SVD, which is dominated by an expensive rank-selection optimization, and ZS-SVD also achieves better perplexity. ZS-SVD takes longer than SVD-LLM because it computes first-order loss-change estimates to guide global selection, but this added cost results in consistently better perplexity than both baselines. All timings are measured on an NVIDIA RTX PRO 6000 Blackwell Max-Q GPU Workstation with 96 GB VRAM.

**Correction validation-loss trend.** Figure 4 tracks the validation loss across alternating truncation ($T_i$) and one-step correction ($C_i$) steps. It trends downward from 3.81 ($T_1$) to 3.02 ($T_6$): each correction lowers the loss, and because the correction gradient has low effective rank, the update increases the rank only marginally, so the subsequent re-truncation incurs only a minor loss increase (Sec. 4.3).

**Distribution of dropped-component $\Delta\mathcal{L}$.** Figure 5 compares the $\Delta\mathcal{L}$ of each dropped singular value for ZS-SVD and SVD-LLM at 20% and 40% pruning. Both distributions are centered near zero, but ZS-SVD is far more concentrated, with roughly half the standard deviation of SVD-LLM at each ratio. SVD-LLM places more mass in the tails, so it drops more components whose removal perturbs the loss strongly. This gap persists under heavier compression, indicating that zero-sum selection favors directions with smaller loss sensitivity, consistent with its lower perplexity.

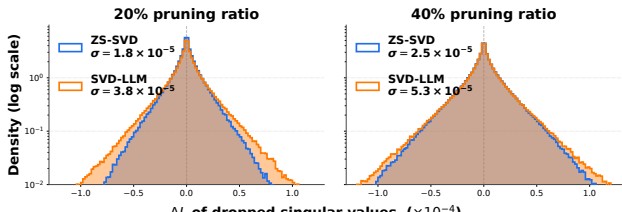

*Figure 5.* Density (log scale) of the predicted loss change $\Delta\mathcal{L}$ for the singular values dropped by ZS-SVD vs. SVD-LLM on LLaMA-7B, at 20% and 40% pruning.

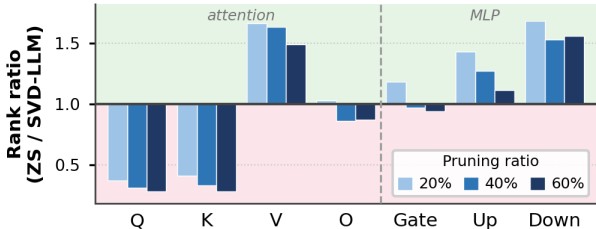

*Figure 6.* Per-module rank ratio (ZS-SVD / SVD-LLM) on LLaMA-7B at 20%, 40%, and 60% pruning.

**Per-module rank allocation.** Figure 6 reports the per-module ratio of ranks retained by ZS-SVD relative to the homogeneous rank allocation of SVD-LLM, across attention (Q, K, V, O) and MLP (Gate, Up, Down) projections. ZS-SVD prunes the query and key projections more aggressively than SVD-LLM's homogeneous allocation (ratio below 1). Conversely, on the MLP projections it retains more rank than the homogeneous allocation (ratio above 1).

## 6. Conclusion

We introduced *Zero-Sum SVD (ZS-SVD)*, a post-training, SVD-based compression framework for LLMs that performs *global* singular-component selection under a shared budget by combining truncation-aware whitening with *signed* first-order calibration-loss estimates in whitened coordinates. By scoring each singular value by its directional loss sensitivity and applying a greedy *zero-sum* selection rule that keeps the cumulative predicted loss drift near zero, ZS-SVD automatically induces heterogeneous ranks across layers without solving an expensive rank-allocation optimization. We also propose an optional truncate–correct–re-truncate step: a single projected-gradient update that briefly leaves the low-rank manifold to recover calibration loss, then re-truncates to the target rank; empirically, gradients are low effective rank, so the re-projection error is small. Across multiple LLM families and scales, ZS-SVD consistently improves perplexity and zero-shot accuracy over strong SVD baselines (e.g., ASVD, SVD-LLM, Dobi-SVD) and structured pruning methods across a range of compression ratios, while also delivering meaningful inference speedups and favorable truncation time compared to optimization-heavy alternatives.

## Acknowledgements

We acknowledge support from Lambda Labs through a Lambda Cloud Research Credit award and from NVIDIA through the NVIDIA Academic Grant Program. This research was also partially supported by the Institute of Education Sciences, U.S. Department of Education, through Grant R305C240010, and by NSF CAREER Award No. 2339898. The opinions expressed are those of the authors and do not necessarily represent the views of the Institute of Education Sciences, the U.S. Department of Education, or the National Science Foundation.

## Impact Statement

This paper presents a post-training compression method for large language models based on low-rank SVD truncation guided by calibration-loss sensitivity. The primary intended impact is to reduce the memory footprint and inference cost of deploying neural language models, which can lower energy consumption and improve accessibility on resource-constrained hardware.

Like other advances in model efficiency, our method could also enable broader deployment of capable language models, including in settings where they may be misused (e.g., for generating misleading or harmful content) or where compressed models inherit and potentially amplify biases present in the underlying pretrained models and calibration data. Our approach does not introduce new data sources, does not change the model's training objective, and does not provide mechanisms for content filtering or bias mitigation; responsible deployment therefore remains essential. We encourage practitioners to evaluate compressed models for reliability, bias, and safety on their target domains and to follow established best practices for monitoring and governance when deploying language technologies.

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

# Appendix

## A. Proofs

**Theorem A.1** (Whitened truncation yields singular value reconstruction loss). *Assume $SS^\top = XX^\top$ and let $A = WS = U\Sigma V^\top$. Let $A_k$ be the rank-$k$ truncation of $A$ and $W'_k = A_k S^{-1}$. Then*

$$\|WX - W'_k X\|_F^2 = \sum_{i>k} \sigma_i^2. \tag{15}$$

*Proof of Theorem A.1.* Using the identity $\|M\|_F^2 = \operatorname{tr}(MM^\top)$, we expand

$$\|WX - W'_k X\|_F^2 = \operatorname{tr}\Big((W - W'_k)XX^\top(W - W'_k)^\top\Big). \tag{16}$$

Under the assumption $SS^\top = XX^\top$, this becomes

$$\|WX - W'_k X\|_F^2 = \operatorname{tr}\Big((W - W'_k)SS^\top(W - W'_k)^\top\Big) \tag{17}$$

$$= \|(W - W'_k)S\|_F^2. \tag{18}$$

By definition, $A = WS$ and $W'_k = A_k S^{-1}$, hence

$$(W - W'_k)S = WS - A_k S^{-1}S = A - A_k. \tag{19}$$

Therefore,

$$\|WX - W'_k X\|_F^2 = \|A - A_k\|_F^2. \tag{20}$$

Let $A = U\Sigma V^\top$ be the SVD of $A$ and let $A_k = U_k \Sigma_k V_k^\top$ be its rank-$k$ truncation. Then

$$A - A_k = \sum_{i>k} \sigma_i\, u_i v_i^\top, \tag{21}$$

and since $\{u_i v_i^\top\}$ are orthonormal in the Frobenius inner product,

$$\|A - A_k\|_F^2 = \sum_{i>k} \sigma_i^2. \tag{22}$$

Combining the displays yields (15). □

**Corollary A.2** (Optimality of truncated SVD in the whitened space). *Under the assumptions of Theorem A.1, $W'_k$ minimizes $\|WX - W'X\|_F$ over all rank $k$ matrices $W'$. Equivalently, truncating the smallest singular values of $A = WS$ is optimal for activation reconstruction at rank $k$.*

*Proof of Corollary A.2.* Assume the conditions of Theorem A.1, and additionally that $S$ is invertible (as in practice,

---

**Algorithm 1** Initialization for global zero sum selection

> **Input:** weights $\{W_\ell \in \mathbb{R}^{m_\ell \times n_\ell}\}_{\ell=1}^L$, activations $\{X_\ell\}_{\ell=1}^L$, retention rate $\rho \in (0,1]$
> **Output:** tuples $\{(U_\ell, \Sigma_\ell, V_\ell, S_\ell, \pi_\ell, p_\ell, k_{\text{thr},\ell})\}_{\ell=1}^L$, heaps $\mathcal{Q}_+, \mathcal{Q}_-$, costs $\{\text{cost}(\ell)\}_{\ell=1}^L$, budget $B \leftarrow (1-\rho)\sum_{\ell=1}^L m_\ell n_\ell$
> Initialize empty min heaps $\mathcal{Q}_+ \leftarrow \emptyset$, $\mathcal{Q}_- \leftarrow \emptyset$
> **for** $\ell = 1$ **to** $L$ **do**
>     Set $k_{\text{thr},\ell} \leftarrow \left\lceil \frac{m_\ell n_\ell}{m_\ell + n_\ell} \right\rceil$
>     Initialize pointer $p_\ell \leftarrow 1$
>     Set $\text{cost}(\ell) \leftarrow 0$      // no savings until $k \le k_{\text{thr},\ell}$
>     Estimate $S_\ell$ from $X_\ell$ so that $S_\ell S_\ell^\top \approx X_\ell X_\ell^\top$
>     Form $A_\ell = W_\ell S_\ell$ and compute SVD $A_\ell = U_\ell \Sigma_\ell V_\ell^\top$
>     Compute whitened gradient $H_\ell \leftarrow (\nabla_{W_\ell}\mathcal{L})S_\ell^{-\top}$
>     Compute $g_{\sigma,\ell} \leftarrow \operatorname{diag}(U_\ell^\top H_\ell V_\ell)$
>     For each $i$: set $\Delta\mathcal{L}_{\ell,i} \leftarrow -\sigma_{\ell,i}\, g_{\sigma,\ell,i}$
>     Sort indices $\pi_\ell$ so that $\sigma_{\ell,\pi_\ell(1)} \le \cdots \le \sigma_{\ell,\pi_\ell(r_\ell)}$
>     $i \leftarrow \pi_\ell(p_\ell)$, $\Delta \leftarrow \Delta\mathcal{L}_{\ell,i}$
>     Push $(\ell, i, \Delta)$ into $\mathcal{Q}_+$ if $\Delta \ge 0$, else into $\mathcal{Q}_-$
> **end for**

this is ensured by using $C + \lambda I$ with $\lambda > 0$). For any rank-$k$ matrix $W'$, define $B \triangleq W'S$. Then $\operatorname{rank}(B) \le k$ and

$$\|WX - W'X\|_F^2 = \|(W - W')S\|_F^2 = \|A - B\|_F^2. \tag{23}$$

Conversely, for any rank-$k$ matrix $B$, setting $W' = BS^{-1}$ yields $\operatorname{rank}(W') \le k$ and $\|A - B\|_F = \|(W - W')S\|_F$. Hence minimizing $\|WX - W'X\|_F$ over rank-$k$ matrices $W'$ is equivalent to minimizing $\|A - B\|_F$ over rank-$k$ matrices $B$.

By the Eckart–Young–Mirsky theorem, the rank-$k$ truncated SVD $A_k$ uniquely minimizes $\|A - B\|_F$ over all rank-$k$ matrices $B$, and the minimum value is $\|A - A_k\|_F^2 = \sum_{i>k} \sigma_i^2$. Taking $B = A_k$ and mapping back via $W'_k = A_k S^{-1}$ gives the stated optimality. □

## B. ZS-SVD algorithm

**Low-rank thresholding and budget accounting.** A rank-$k$ factorization of an $m \times n$ weight stores $k(m+n)$ parameters, while the dense matrix stores $mn$. Low-rank becomes storage-saving only when $k(m+n) \le mn$, i.e.,

$$k_{\text{thr}} = \frac{mn}{m+n}. \tag{24}$$

We incorporate this into the global budget as follows. While a layer's current rank $k_\ell$ remains above $k_{\text{thr},\ell}$, switching to low-rank would not reduce storage, so dropping additional singular components does not contribute to the parameter-removal budget and we set its per-drop cost to 0. Once $k_\ell \le k_{\text{thr},\ell}$, each further drop reduces the stored factors by

one column/row and saves $(m_\ell + n_\ell)$ parameters, so we set $\text{cost}(\ell) = m_\ell + n_\ell$ thereafter. Finally, after selection terminates, if a layer's final rank satisfies $k_\ell > k_{\text{thr},\ell}$, we keep the original dense weight $W_\ell$ (no low-rank replacement) to avoid introducing truncation noise when factorization is not worthwhile; otherwise we store the truncated factors.

---

**Algorithm 2** Global zero sum selection and reconstruction

---

**Input:** $\{(U_\ell, \Sigma_\ell, V_\ell, S_\ell, \pi_\ell, p_\ell, k_{\text{thr},\ell})\}_{\ell=1}^L$, heaps $\mathcal{Q}_+, \mathcal{Q}_-$, budget $B$, costs $\{\text{cost}(\ell)\}_{\ell=1}^L$
**Output:** compressed weights $\{W_\ell'\}_{\ell=1}^L$ (dense or factored)
Initialize removed budget $b \leftarrow 0$ and running sum $s \leftarrow 0$
**while** $b < B$ **and** $(\mathcal{Q}_+ \neq \emptyset$ **or** $\mathcal{Q}_- \neq \emptyset)$ **do**
    Prefer $\mathcal{Q}_+$ if $s \leq 0$, otherwise prefer $\mathcal{Q}_-$
    **if** $s \leq 0$ **then**
        Pop $(\ell, i, \Delta)$ from $\mathcal{Q}_+$, else pop from $\mathcal{Q}_-$ if $\mathcal{Q}_+$ is empty
    **else**
        Pop $(\ell, i, \Delta)$ from $\mathcal{Q}_-$, else pop from $\mathcal{Q}_+$ if $\mathcal{Q}_-$ is empty
    **end if**
    Mark $(\ell, i)$ as removed and update $s \leftarrow s + \Delta$
    Advance $p_\ell \leftarrow p_\ell + 1$      // one more component removed from layer $\ell$
    Let $k_\ell \leftarrow \min(m_\ell, n_\ell) - (p_\ell - 1)$     // remaining components
    **if** $k_\ell \leq k_{\text{thr},\ell}$ **then**
        Set $\text{cost}(\ell) \leftarrow m_\ell + n_\ell$     // savings per further removal
    **else**
        Set $\text{cost}(\ell) \leftarrow 0$
    **end if**
    Update budget $b \leftarrow b + \text{cost}(\ell)$
    **if** $p_\ell \leq \min(m_\ell, n_\ell)$ **then**
        $j \leftarrow \pi_\ell(p_\ell)$, $\Delta' \leftarrow \Delta\mathcal{L}_{\ell,j}$
        Push $(\ell, j, \Delta')$ into $\mathcal{Q}_+$ if $\Delta' \geq 0$, else into $\mathcal{Q}_-$
    **end if**
**end while**
**for** $\ell = 1$ **to** $L$ **do**
    Let $k_\ell \leftarrow \min(m_\ell, n_\ell) - (p_\ell - 1)$
    **if** $k_\ell > k_{\text{thr},\ell}$ **then**
        Set $W_\ell' \leftarrow W_\ell$     // keep dense (skip truncation)
    **else**
        Let $\Sigma_\ell'$ be $\Sigma_\ell$ with removed components zeroed
        Set $W_{u,\ell}' \leftarrow U_\ell(\Sigma_\ell')^{1/2}$
        Set $W_{v,\ell}' \leftarrow (\Sigma_\ell')^{1/2}V_\ell^\top S_\ell^{-1}$
        Set $W_\ell' \leftarrow W_{u,\ell}'W_{v,\ell}'$
    **end if**
**end for**

---

*Table 11.* Ablation of correction variants on LLaMA-7B, Wiki-Text2.

| | $\alpha$-BLEND | | | GD CORR. | | | PROJ. | PROJ |
|---|---|---|---|---|---|---|---|---|
| | 0.25 | 0.50 | 0.75 | $\eta{=}10^{-2}$ | $\eta{=}10^{-3}$ | $\eta{=}10^{-4}$ | $\Delta$ | GRAD |
| PPL↓ | 42.2 | 42.8 | 53.8 | 40.4 | 38.3 | 47.7 | 48.8 | 26.9 |

### B.1. Ablation: correction variants

Table 11 ablates several correction strategies applied after the first truncation stage. Let $W \in \mathbb{R}^{m \times n}$ denote the original (uncompressed) weight, and let $W_k'$ be its rank-$k$ truncation after the first stage. We evaluate each strategy by applying a single correction update to $W_k'$ using the same calibration loss $\mathcal{L}$, followed by re-truncation back to rank $k$.

$\alpha$-**blend.** We linearly interpolate between the truncated weights and the original weights,

$$W_\alpha = (1 - \alpha)W_k' + \alpha W, \tag{25}$$

and then re-truncate $W_\alpha$ back to rank $k$. We sweep $\alpha \in \{0.25, 0.50, 0.75\}$.

**GD correction.** We apply a single gradient descent step at the truncated point,

$$W^+ = W_k' - \eta g, \qquad g \triangleq \nabla_W \mathcal{L}(W_k'), \tag{26}$$

and then re-truncate $W^+$ to rank $k$. This update does not use information from the teacher weights $W$.

**Projection-based corrections.** Define the truncation residual (teacher residual)

$$\Delta W \triangleq W - W_k'. \tag{27}$$

We compare two complementary projection directions. *Proj. $\Delta$* projects the gradient onto the residual direction,

$$g_\Delta \triangleq \frac{\langle g, \Delta W \rangle}{\langle \Delta W, \Delta W \rangle} \Delta W, \qquad W^+ = W_k' + g_\Delta, \tag{28}$$

followed by re-truncation to rank $k$. In contrast, *Proj. Grad* (ours) uses the one-step correction derived in Sec. 4.3, projecting the residual onto the gradient direction,

$$\Delta W' \triangleq \frac{\langle g, \Delta W \rangle}{\langle g, g \rangle} g, \qquad W^+ = W_k' + \Delta W', \tag{29}$$

followed by re-truncation to rank $k$.

*Table 12.* Comparison with SVD-LLMv2 at 20% pruning (WikiText-2 PPL ↓, average zero-shot accuracy ↑).

| MODEL | METHOD | PPL ↓ | AVG. ACC ↑ |
|---|---|---|---|
| LLaMA-7B | SVD-LLMv2 | 7.12 | **0.54** |
| | ZS-SVD (OURS) | **6.74** | 0.53 |
| OPT-6.7B | SVD-LLMv2 | 13.46 | 0.49 |
| | ZS-SVD (OURS) | **11.40** | **0.51** |

*Table 13.* Correction stability at pruning ratio 0.4 on LLaMA-7B (WikiText-2 PPL ↓), over 3 runs; each step draws a fresh batch of 4 samples.

| CORRECTION STEP | MEAN | STD |
|---|---|---|
| 1 | 9.93 | 0.07 |
| 2 | 9.78 | 0.05 |
| 3 | 9.52 | 0.06 |
| 4 | 9.41 | 0.04 |
| 5 | 9.33 | 0.11 |

*Table 14.* Fixed vs. per-stage resampled calibration on LLaMA-7B at 60% pruning (end-of-stage WikiText-2 PPL ↓).

| STAGE | FIXED | RESAMPLED |
|---|---|---|
| 1 | 45.17 | 45.17 |
| 2 | **26.92** | 27.03 |
| 3 | **23.65** | 23.67 |
| 4 | 23.33 | **22.54** |
| 5 | 22.10 | **20.97** |
| 6 | 20.41 | **20.05** |
| 7 | 20.63 | **19.58** |

## C. Comparison with SVD-LLMv2

We compare against SVD-LLMv2 (Wang et al., 2025a) using the numbers reported in their paper, at the 20% pruning ratio they provide. Table 12 reports WikiText-2 perplexity and average zero-shot accuracy at 20% pruning. On LLaMA-7B, ZS-SVD improves perplexity over SVD-LLMv2 with comparable accuracy (a 0.01 gap); on OPT-6.7B, ZS-SVD improves both perplexity and accuracy.

## D. Sensitivity analysis

**Correction batch size.** Each correction step uses only 4 randomly sampled calibration examples, whereas singular-value profiling uses all 256 WikiText samples. To assess stability, we run 3 independent trials at pruning ratio 0.4 and report the mean and standard deviation of perplexity after each correction step, starting from the post-truncation perplexity of 11.44 and drawing a fresh batch of 4 samples per step (Table 13). Perplexity decreases steadily with more correction steps and the standard deviation stays small throughout, indicating that the small correction batch is both efficient and stable.

**Calibration data across stages.** For fair comparison with prior methods, we use the same fixed 256 WikiText calibration samples across all compression stages, which also avoids introducing new information between stages. We additionally test resampling 256 fresh WikiText samples at each truncation–correction stage. Table 14 reports the end-of-stage perplexity for both strategies on LLaMA-7B at 60% pruning. Resampling tends to improve later-stage perplexity, suggesting that fresh information across stages is helpful.

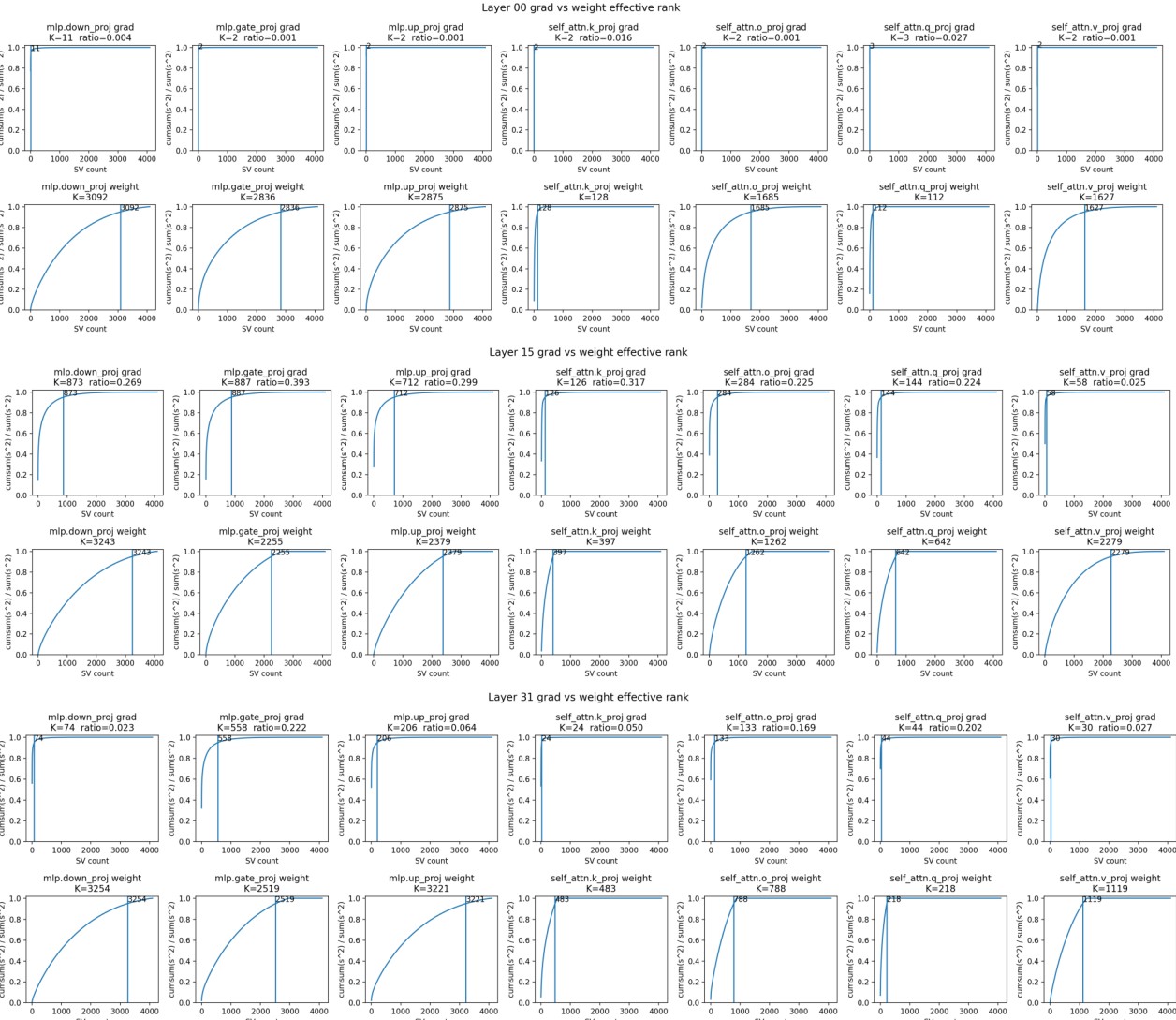

*Figure 7.* Gradient vs. weight effective rank at Layers 1, 16, and 32 of LLaMA-2-7B under 20% pruning. For each module, we compute the singular spectrum of the truncated weight $W'$ and the gradient $G = \nabla_W \mathcal{L}(W')$, and define the effective rank using a spectral energy cutoff of 0.95 (vertical dashed line), i.e., the smallest $k$ such that $\sum_{i \le k} \sigma_i^2 / \sum_i \sigma_i^2 \ge 0.95$.

