# OpenReview forum: "Zero Sum SVD: Balancing Loss Sensitivity for Low Rank LLM Compression"
_ICML.cc/2026/Conference — ICML 2026 regular_

### Official Review · Reviewer_MPkG · 2026-03-05

**Soundness:** 2
**Presentation:** 3
**Significance:** 3
**Originality:** 3
**Overall Recommendation:** 4
**Confidence:** 3

**Summary:**

This paper proposes a post-training compression strategy for LLMs based on SVD under a fixed budget. More specifically, a global strategy is used to find the optimal rank of specific matrices to minimize the activation reconstruction error, targeting at a zero sum on the final loss. Also, the authors derived the equivalence between activation reconstruction error and whitened truncation error. Authors also propose a correction step based on the observation that gradients are usually low-rank. Comprehensive experiments demonstrate the effectiveness of the proposed method on LLMs of various sizes and different datasets.

**Compliance With Llm Reviewing Policy:**

Affirmed.

**Final Justification:**

The rebuttal has addressed my main concerns, so I would like to maintain a score of 4, which signifies acceptance. The paper should include the statistics presented during the rebuttal stage in the final version for an improved soundess of the proposed method. Also, the empirical evidience for Theorem 3.1 should be included to help support its theoretical derivation. This paper provides a solid contribution to low-rank LLM compression.

**Key Questions For Authors:**

Questions:
1. Is the Delta L usually positive or negative when dropping a singular value?

**Limitations:**

yes

**Strengths And Weaknesses:**

Strengths:
1. The motivation behind the zero-sum selection strategy, which moves beyond the approximation error of weight matrices, is clear and intuitive. This then further motivates the derivation of Theorem 3.1.
2. The paper is well-written and easy-to-follow.
3. Comprehensive experiments and ablation studies demonstrate the effectiveness of the proposed method on LLMs of various sizes and different datasets.

Weaknesses:
1. The zero-sum strategy should be justified by plotting the histogram of Delta L. Currently, it’s not possible to gauge the amplitudes and directions of the changes in Delta L. This is needed to support the proposed method, as truncation of singular values likely lead to increases in the loss values.
2. Theorem 3.1 should be supported with some emperical results.
3. Best and second-best performances should be highlighted in the tables for better readability

---

> ### Author Rebuttal · Authors · 2026-03-31
>
> We thank the reviewer for the constructive feedback and address each point below.
>
> ---
>
> ## W1: Distribution of $\Delta L$
>
> We agree that visualizing the distribution of $\Delta L$ would strengthen the motivation for the zero-sum strategy, and we will add a histogram in the final version. Using the uncompressed LLaMA-7B model and the same WikiText calibration set, we computed $\Delta L$ for every singular value and summarize the distribution below over all layers.
>
> | | 25th pct. | 50th pct. | 75th pct. | Mean | Std |
> |---|---|---|---|---|---|
> | $\Delta L$ | $-1.75\times10^{-5}$ | $-3.26\times10^{-8}$ | $1.67\times10^{-5}$ | $3.87\times10^{-7}$ | $4.00\times10^{-4}$ |
>
> We also compare the |ΔL| values of the pruned singular values selected by ZS-SVD and SVD-LLM. ZS-SVD consistently prunes components with smaller |ΔL| across the reported percentiles, and the gap widens in the tail; at the maximum, SVD-LLM's worst-case pruned |ΔL| is about 16x larger. We observe the same pattern at 40% pruning.
>
> **Pruning Ratio: 20%**
>
> | Percentile | ZS-SVD \|ΔL\| | SVD-LLM \|ΔL\| | Ratio (ZS / SVD-LLM) |
> |---|---|---|---|
> | 1% | 1.41e-08 | 1.88e-08 | 0.75 |
> | 10% | 6.51e-07 | 7.86e-07 | 0.83 |
> | 25% | 2.31e-06 | 2.75e-06 | 0.84 |
> | 50% | 6.77e-06 | 8.22e-06 | 0.82 |
> | 90% | 2.89e-05 | 3.79e-05 | 0.76 |
> | 99% | 6.07e-05 | 9.31e-05 | 0.65 |
> | 99.9% | 8.76e-05 | 3.84e-04 | 0.23 |
> | 100% | 2.49e-04 | 3.92e-03 | 0.06 |
>
> ---
>
> ## W2: Empirical Results for Theorem 3.1
>
> Theorem 3.1 is supported by a formal analytical proof, and we provide empirical validation here. For a given layer, the theorem states that the reconstruction MSE of WX, after rank-k truncation equals the sum of the squared pruned singular values of the whitened weight matrix. To verify this, we compare two quantities: first, the average reconstruction error after replacing the original weight matrix with its rank-k truncated version; second, the average sum of the squared singular values removed by truncation. By the theorem, these two quantities should be equal. The mean relative error across module types confirms this closely, and the small nonzero discrepancy is due to numerical approximation in the Cholesky decomposition and FP32-to-FP16 casting, rather than a theoretical inconsistency. We will include the full detailed proof in the supplementary material of the final version.
>
> **ZS-SVD, Pruning Ratio = 0.6**
>
> | Module | Mean ‖WX − W'X‖² | Mean Σσ² | Mean Rel. Error |
> |---|---|---|---|
> | q_proj | 668.909 | 669.624 | 1.05e-03 |
> | k_proj | 737.256 | 738.053 | 1.04e-03 |
> | v_proj | 560.679 | 561.271 | 1.08e-03 |
> | o_proj | 48.943 | 48.997 | 1.09e-03 |
> | gate_proj | 498.398 | 499.077 | 1.37e-03 |
> | up_proj | 406.292 | 406.836 | 1.32e-03 |
> | down_proj | 122.015 | 122.146 | 1.18e-03 |
> | **All** | **433.193** | **433.696** | **1.17e-03** |
>
>
> ---
>
> ## W3: Highlighting the Best Performers
>
> We will highlight the best and second-best results in all tables in the final version.
>
> ---
>
> ## Q1: Sign of $\Delta L$, Summary Statistics
>
> We provide the summary statistics below and will include both these numbers and a histogram of $\Delta L$ in the final version.
>
> | | Mean | Std | Count |
> |---|---|---|---|
> | Positive $\Delta L$ | 5.26e-05 | 4.35e-04 | 455,363 (49.63%) |
> | Negative $\Delta L$ | -5.11e-05 | 3.55e-04 | 462,141 (50.37%) |
>
>
> ---
>
> We thank the reviewer again for the thoughtful feedback and hope the additional analyses above further strengthen the paper.

---

> > ### Author Rebuttal · Reviewer_MPkG · 2026-04-02
> >
> > Thank you for the detailed response. My concerns have been adequately addressed. Please make sure to include these statistics and a histogram of Delta L in the final version, which confirm the validity of theorem 3.1 and supports the motivation for the zero-sum strategy. I'll maintain the score as it already signifies acceptance.

---

> > > ### Author Response · Authors · 2026-04-02
> > >
> > > We sincerely thank the reviewer for the thoughtful feedback and for acknowledging that the concerns have been addressed. We appreciate the helpful suggestions, and we will include the requested statistics and the histogram of ΔL in the final version.

---

### Official Review · Reviewer_LMRw · 2026-03-12

**Soundness:** 2
**Presentation:** 1
**Significance:** 2
**Originality:** 2
**Overall Recommendation:** 3
**Confidence:** 5

**Summary:**

This paper proposes Zero-Sum SVD (ZS-SVD), a post-training low-rank compression method for LLM. It follows the activation-aware/whitening SVD framework, but instead of simply truncating each matrix locally, it estimates a first-order sensitivity score associated with the calibration loss for each singular value, and then performs signed global singular component selection across the entire model.

**Compliance With Llm Reviewing Policy:**

Affirmed.

**Key Questions For Authors:**

1. The presentation of experimental results is poor; the authors should have bolded the results that significantly outperform existing methods. I also noticed that SVD-LLMv2 was not compared.

2. The paper uses the "select the smallest ΔL from the positive and negative candidate piles based on the current cumulative drift sign" rule for zero-sum selection. This seems like a greedy heuristic, not an algorithm with guaranteed optimality. Is ZS-SVD implicitly solving rank allocation, or is it simply performing finer-grained component selection?

3. Is the first-order loss sensitivity reliable at high compression rates? The paper doesn't provide higher-order analyses, such as Hessian.

**Limitations:**

yes

**Strengths And Weaknesses:**

Strengths:
1. ZS-SVD introduces singular-value-level signed loss sensitivity into global selection.

2. ZS-SVD can automatically generate heterogeneous ranks without requiring expensive layer optimization like Dobi-SVD. Under the same settings, ZS-SVD has a truncation time of 15.9 minutes, while Dobi-SVD has a truncation time of 19.25 hours.

3. In the LLaMA-7B/Vicuna-7B comparison with 30% pruning, ZS-SVD outperforms ASVD, FWSVD, SVD-LLM, and Dip-SVD listed in the paper in both PPL and average accuracy.

---

> ### Author Rebuttal · Authors · 2026-03-31
>
> We thank the reviewer for the constructive feedback. We address each question below.
>
> ---
>
> ## Q1: Presentation: Missing Bold Results and No SVD-LLMv2 Comparison
>
> **Bold formatting.** We will highlight the best and second-best results in all tables in the final version.
>
> **SVD-LLMv2 comparison.** SVD-LLMv2, which is already cited in our paper, did not have a public implementation available at the time of submission; the released repository only contains the SVD-LLM v1 code. Nevertheless, we compare against the results reported in their paper at the pruning ratios they provide. The comparison is shown below.
>
> **LLaMA-7B (Pruning Ratio: 20%)**
>
> | Method | PPL WikiText-2 (↓) | Avg. Accuracy (↑) |
> |---|---|---|
> | SVD-LLMv2 | 7.12 | **0.54** |
> | **ZS-SVD (Ours)** | **6.74** | 0.53 |
>
> **OPT-6.7B (Pruning Ratio: 20%)**
>
> | Method | PPL WikiText-2 (↓) | Avg. Accuracy (↑) |
> |---|---|---|
> | SVD-LLMv2 | 13.46 | 0.49 |
> | **ZS-SVD (Ours)** | **11.40** | **0.51** |
>
> Since SVD-LLMv2 reports results mainly at the 20% pruning ratio, a broader comparison at higher compression levels was not possible. On LLaMA-7B, ZS-SVD improves PPL over SVD-LLMv2, while average accuracy is comparable with a 0.01 gap. On OPT-6.7B, ZS-SVD improves both PPL and average accuracy. We will add these results to the paper.
>
> ---
>
> ## Q2: Greedy Heuristic vs. Principled Algorithm
>
> We agree that the zero-sum rule is a greedy heuristic rather than an algorithm with a theoretical optimality guarantee. We do not claim otherwise in the paper. More precisely, ZS-SVD performs global singular-value selection under a fixed pruning budget in the whitened space. This is finer-grained than assigning one rank per layer in advance, and it implicitly induces a heterogeneous rank allocation across layers, but it does not explicitly solve a rank-allocation optimization problem.
>
> A natural comparison is DOBI-SVD, which follows the more explicit route suggested by the reviewer: it formulates rank allocation as an end-to-end differentiable optimization targeting perplexity. While principled, this comes with substantially higher cost. As shown in Table 8, DOBI-SVD requires 19.25 hours, whereas ZS-SVD finishes in 15.9 minutes and achieves better perplexity. We believe this shows that a simple heuristic can still offer a strong practical tradeoff between quality and efficiency.
>
> ---
>
> ## Q3: Reliability of First-Order Sensitivity at High Compression Rates
>
> **On compression ratios.** The pruning ratios we study are standard in the SVD compression literature. Even at the highest commonly used ratio of 60%, ZS-SVD continues to outperform competing baselines. To further stress-test the method, we also report results at 80% pruning:
>
> | Method | Compression | PPL (WikiText-2) |
> |---|---|---|
> | ASVD | 80% | 80425 |
> | FWSVD | 80% | 96872 |
> | SVD-LLM | 80% | 1349 |
> | **ZS-SVD (Ours)** | 80% | 332.20 |
>
> ZS-SVD still outperforms the baselines even at 80% pruning.
>
> **On the Hessian concern.** This is a fair point, and we agree that our method relies on a first-order approximation. In principle, second-order information could be incorporated, but the main challenge is computational cost: the full Hessian is a $d \times d$ matrix, where $d$ is the number of parameters, making exact computation infeasible for LLMs.
>
> To partially address this, we evaluate the diagonal Fisher information matrix as a lightweight approximation to second-order curvature. The results below cover 20%, 40%, and 60% pruning ratios:
>
> | Method | Compression | PPL (WikiText-2) |
> |---|---|---|
> | **ZS-SVD (First-order)** | 20% | **6.74** |
> | ZS-SVD (Diagonal Fisher) | 20% | 6.83 |
> | **ZS-SVD (First-order)** | 40% | **11.44** |
> | ZS-SVD (Diagonal Fisher) | 40% | 11.98 |
> | **ZS-SVD (First-order)** | 60% | **45.17** |
> | ZS-SVD (Diagonal Fisher) | 60% | 46.32 |
>
> The first-order approximation consistently outperforms the diagonal Fisher variant. We believe this is because diagonal Fisher remains a coarse approximation that ignores off-diagonal curvature. This suggests that our first-order criterion is effective in practice relative to this lightweight second-order proxy. More faithful approximations, such as blockwise or Kronecker-factored methods, are a natural direction for future work, but would require substantially higher computational cost.
>
> ---
>
> We hope these clarifications and additional experiments address the reviewer’s concerns, and we would greatly appreciate it if they are taken into account in the final evaluation.

---

> > ### Author Rebuttal · Reviewer_LMRw · 2026-04-01
> >
> > Thanks to the authors for their efforts. There are many second-order approximation methods, and while Fisher's may not be optimal for ZS-SVD, it is sufficient. Based on the current rebuttals, the experimental results seem acceptable.

---

> > > ### Author Response · Authors · 2026-04-02
> > >
> > > We sincerely thank the reviewer for the thoughtful feedback and for acknowledging that the concerns have been addressed. We are glad the additional experiments and clarifications were helpful. We would be grateful if this is reflected in the updated assessment, should the reviewer feel the concerns are now fully resolved.

---

### Official Review · Reviewer_QP6V · 2026-03-13

**Soundness:** 3
**Presentation:** 3
**Significance:** 3
**Originality:** 3
**Overall Recommendation:** 5
**Confidence:** 5

**Summary:**

This paper introduced a post-training low-rank compression technique, called Zero-Sum SVD (ZS-SVD). Each weight is whitened by a Cholesky factor of a feature matrix $X^T X$, then for each singular value of the whitened weight, a first-order sensitivity of the loss function is computed, which is either positive or negative, where a negative sensitivity indicates removing the singular value decreases the calibration loss. Based on the sensitivity, ZS-SVD prunes the singular values across the layers by matching the cumulative sum of the loss sensitivities to near zero.  Additionally, a weight is updated with one-step gradient descent followed by re-truncation. The proposed method outperforms strong post-training low-rank compression baselines in LLM compression.

**Compliance With Llm Reviewing Policy:**

Affirmed.

**Final Justification:**

The rebuttal addressed all of my concerns. I keep my score and confidence the same (5/5).

**Key Questions For Authors:**

1. In Table 6, why does the “most negative” strategy perform poorly? Is it because the spectrums are overfit to the calibration set?
2. In Table 1, at each iteration of ZS-SVD, is the same set of calibration data used? If so, what happens if a new set of calibration data is provided?
3. Was the throughput comparison performed with the compiled model (e.g., with `torch.compile` )?

**Limitations:**

Yes

**Strengths And Weaknesses:**

- Soundness
    - The theories stated in the paper seem sound. The proposed method is examined thoroughly with various models and benchmarks.
- Presentation
    - The paper is well structured, and the proposed method is clearly explained.
- Significance
    - The paper advances both the theoretical framework and the compression performance of low-rank LLM compression.
- Originality
    - The singular value sensitivity and the zero-sum global truncation method are novel and original. In particular, the singular value sensitivity made further progress from the ideas of whitening (SVD-LLM) and gradient descent on singular values (Dobi-SVD).

Some minor weaknesses are stated below:

1. It is unclear why the ZS-SVD achieves the highest throughput compared to SVD-LLM and Dobi-SVD. Although the layer-wise compression ratio can be different from method to method, all are compressed into a low-rank by the same average compression ratio. It would be clearer if the text provides what factors of ZS-SVD contribute to the enhanced throughput from the other two methods.

---

> ### Author Rebuttal · Authors · 2026-03-31
>
> We thank the reviewer for the insightful comments. We address each point below.
>
> ---
>
> ## W1: Throughput Analysis
>
> This is an interesting point. We performed further analysis to investigate this, and the results are presented below.
>
> We find that ZS-SVD redistributes capacity in a structured way across module types. Specifically, it prunes Q/K more aggressively while preserving V and down projections, a pattern that emerges automatically. The table below shows the mean rank ratio (ZS-SVD / SVD-LLM) per module type at two pruning ratios on LLaMA-7B:
>
> **Pruning Ratio: 40%**
>
> | Module | SVD-LLM Rank | Ours Mean Rank | Ratio |
> |---|---|---|---|
> | q_proj | 1229 | 386 | **0.31** |
> | k_proj | 1229 | 410 | **0.33** |
> | v_proj | 1229 | 1999 | **1.63** |
> | o_proj | 1229 | 1055 | 0.86 |
> | gate_proj | 1791 | 1734 | 0.97 |
> | up_proj | 1791 | 2270 | 1.27 |
> | down_proj | 1791 | 2746 | **1.53** |
>
> Although ZS-SVD and SVD-LLM have the same compressed size, they allocate the budget differently. ZS-SVD prunes attention more aggressively and preserves more MLP capacity. The same pattern appears at 60% pruning. We speculate that one of the reasons for our better throughput is that latency is not determined by parameter count alone. Prior systems work suggests that attention is often more sensitive to memory traffic and IO than highly optimized MLP operations, so reallocating the same low-rank budget away from attention and toward MLP can improve runtime even at matched model size [FlashAttention, Dao et al.].
>
> We believe that fully understanding the throughput differences requires experiments across different sequence lengths, along with meticulous latency profiling of the attention and MLP layers, and we will include this analysis in the final version of the paper.
>
> ---
>
> ## Q1: Most-Negative Selection Strategy
>
> Overfitting to the calibration set is one possible explanation. We also believe several factors are entangled here. As pruning progresses, the model moves farther from the point where the initial ΔL values were estimated, making those estimates less reliable, especially at high compression ratios, where recalibration and recomputation of the ΔL values become increasingly important.
>
> The most-negative strategy makes this worse by aggressively selecting the lowest ΔL values early on, which changes the loss landscape more abruptly and causes the initial estimates to go stale faster. In contrast, the zero-sum strategy keeps cumulative loss drift closer to zero and stays nearer to the original operating point.
>
> Another possible factor is the variation in ΔL scale across layers. A greedy most-negative strategy may overprune a small set of highly sensitive layers, which is especially harmful at high compression ratios.
>
> ---
>
> ## Q2: Calibration Data Across Stages
>
> Yes. For fair comparison with prior methods, we use the same fixed 256 WikiText calibration samples throughout all compression stages. This also avoids introducing new information between stages.
>
> That said, we also experiment with randomly resampling 256 new calibration samples from WikiText at each truncation-correction stage. The results below report perplexity at the end of each stage for both strategies on LLaMA-7B:
>
> **Pruning Ratio: 60%**
>
> | Stage | Fixed Calibration | Resampled per Stage |
> |---|---|---|
> | 1 | 45.17 | 45.17 |
> | 2 | 26.92 | 27.03 |
> | 3 | 23.65 | 23.67 |
> | 4 | 23.33 | 22.54 |
> | 5 | 22.10 | 20.97 |
> | 6 | 20.41 | 20.05 |
> | 7 | 20.63 | 19.58 |
>
> At 60% pruning, resampling the calibration set at each truncation-correction stage tends to improve later-stage performance, suggesting that fresh information across stages is helpful. The same trend holds at 40% pruning, though with smaller gains.
>
> ---
>
> ## Q3: Throughput with `torch.compile`
>
> No, the experiments reported in the paper did not use `torch.compile`. We additionally evaluated the models with and without `torch.compile` on an RTX 5000 (25GB), using sequence length 128 and generation length 128. Results for LLaMA-2 7B at 60% pruning ratio:
>
> | Method | Compile | tok/s | Peak Mem (GB) |
> |---|---|---|---|
> | ZS-SVD | ✗ | 751.9 | 13.4 |
> | ZS-SVD | ✓ | **831.8** | 13.4 |
> | DOBI-SVD | ✗ | 692.9 | 17.1 |
> | DOBI-SVD | ✓ | 678.4 | 17.1 |
>
> Overall, we did not observe a consistent advantage from `torch.compile`. ZS-SVD maintains higher throughput than DOBI-SVD across both settings.
>
> ---
>
> We thank the reviewer again for the thoughtful feedback and hope the additional analyses above further strengthen the paper.

---

> > ### Author Rebuttal · Reviewer_QP6V · 2026-04-02
> >
> > I appreciate the thorough responses. As I already voted for acceptance of the paper, I will keep my score unchanged.

---

> > > ### Author Response · Authors · 2026-04-03
> > >
> > > We thank the reviewer for the thoughtful feedback and for engaging with our rebuttal. We are glad that the main concerns have been addressed and appreciate the careful evaluation.

---

### Official Review · Reviewer_ZYmV · 2026-03-23

**Soundness:** 3
**Presentation:** 3
**Significance:** 3
**Originality:** 3
**Overall Recommendation:** 4
**Confidence:** 3

**Summary:**

This paper proposes Zero-Sum SVD (ZS-SVD), a post-training, activation-aware low-rank compression method for LLMs that heterogeneously allocates ranks across different layers. The key idea is to score the singular values of each whitened weight matrix via gradient-based singular value sensitivity, and select the singular values and corresponding vectors to remove by maintaining the cumulative sum of global predicted loss change close to zero, balancing positive and negative predicted loss changes. An optional one-step projected gradient correction with re-truncation further recovers performance. Across models based on LLaMA, Vicuna, and OPT, as well as various compression ratios, ZS-SVD outperforms strong SVD baselines (ASVD, SVD-LLM/v2-style settings, Dobi-SVD) and various structured pruning methods, improving perplexity and zero-shot accuracy while offering practical throughput gains and shorter truncation times compared to optimization-based methods.

**Compliance With Llm Reviewing Policy:**

Affirmed.

**Final Justification:**

Thank you for the detailed rebuttal, which has addressed my main concerns, so I would like to maintain a score of 4.

**Key Questions For Authors:**

1.	Can additional experiments be provided to further illustrate the actual error drift and performance improvement under the zero sum approximation constraint versus without it?
2.	For fairness under high compression, can you report baselines with both remapping and the HQ scheme, or explain why certain combinations were not applied?
3.	Does the correction step increase loss after re-truncation (e.g., due to misalignment between the gradient and the low-rank manifold)?

**Limitations:**

No, the discussion could be supplemented regarding scenarios where the zero-sum heuristic may fail under high compression ratios. Further discussion could also address the specific mechanisms by which compressed models may exacerbate bias amplification risks, as well as recommended measures for evaluating the reliability, bias, and safety of compressed models in target domains prior to deployment.

**Strengths And Weaknesses:**

Strengths：
1.	Innovation in Methodology: Introduces a principled, signed first-order sensitivity metric for individual singular values in whitened coordinates, bridging activation-aware SVD and loss-aware selection. Proposes a simple yet effective global "zero-sum" selection rule that balances cumulative predicted loss drift while enforcing per-matrix spectral ordering, thereby yielding heterogeneous ranks without solving a global combinatorial allocation, and uses an optional projected gradient correction to recover loss.
2.	Experimental Evaluation is Relatively Comprehensive: Results cover multiple model families/sizes and benchmarks; ablation studies analyze alternative selectors, gradient rank, throughput on different GPUs, and truncation time, and the results demonstrate the effectiveness of the method.

Weaknesses：
Technical Limitations or Concerns: The zero-sum heuristic theoretically may not guarantee that maintaining the cumulative predicted ΔL close to zero correlates with preserving true loss, especially when cross-layer interactions are amplified under high compression. The one-step correction assumes low-rank gradients; while empirically feasible, the method may still be sensitive to the composition of the calibration batch and could benefit from explicit control over the correction magnitude.
Experimental Gaps and Concerns: The joint optimization of truncation and quantization is not clearly explained in some parts. Some baselines (e.g., Dobi-SVD) are compared in settings with remapping enabled, while others are not; the authors propose the HQ scheme but do not apply it uniformly across all high-compression baselines, which complicates strict fairness. The dramatically high perplexity of certain baselines (e.g., ASVD collapse) suggests fragility under high compression; it would be helpful to confirm whether these baselines were emphasized and optimized, or to include additional updated activation-aware methods beyond those mentioned.

---

> ### Author Rebuttal · Authors · 2026-03-31
>
> We thank the reviewer for the feedback and address the points below.
>
> ---
>
> ## W1: Zero-Sum Heuristic Guarantee
>
> We agree that our method does not come with a theoretical guarantee that enforcing a zero-sum constraint on the predicted ΔL will directly preserve the true loss. However, our experiments show that it is a practical and computationally efficient heuristic that consistently outperforms other SVD-based baselines across a range of compression ratios. We provide further details in Q1.
>
> ---
>
> ## W2: Low-Rank Gradient Assumption and Correction Step Sensitivity
>
> Fig 3 provides empirical support for the low-rank grad assumption, and the works cited in Sec 4.3 provide theoretical support under mild assumptions. For correction sensitivity, each step uses only 4 randomly sampled examples, while profiling uses all 256 WikiText calibration samples. Across 3 runs at pruning ratio 0.4, the resulting PPL standard deviation remains low, indicating that this small correction batch is both efficient and stable.
>
> Starting from the same initial perplexity of 11.44 (after the first truncation), the table below shows how PPL evolves across correction steps, where each step uses a freshly resampled batch of 4 calibration samples:
>
> | Correction Num. | Mean | Std |
> |---|---|---|
> | 1 | 9.93 | 0.07 |
> | 2 | 9.78 | 0.05 |
> | 3 | 9.52 | 0.06 |
> | 4 | 9.41 | 0.04 |
> | 5 | 9.33 | 0.11 |
>
> ---
>
> ## W3 & W4: Fairness Under Remapping and HQ Scheme
>
> We address both concerns in detail under Q2.
>
> ---
>
> ## W5: High Perplexity of Certain Baselines
>
> ASVD is known to be less stable at high compression; its original experiments mostly focus on milder regimes. Using the public code and the same 256 WikiText calibration samples as other baselines, we reproduced behavior consistent with what DOBI-SVD and SVD-LLM report. Since perplexity is exponential in loss, modest loss increases at high compression can produce very large PPL values.
>
> ---
>
> ## Q1: Loss Drift of the Zero-Sum
>
>  Our experimental setup helps isolate the contribution of the zero-sum approximation. ZS-SVD and SVD-LLM operate in the same whitened singular-value space; they differ only in how singular values are selected under a fixed pruning budget. Thus, the performance gap isolates the effect of zero-sum selection.
>
> Validation loss and loss drift vs. SVD-LLM on LLaMA-7B (WikiText-2):
>
> | Pruning Ratio | Method | Loss (↓) | Loss Drift (↓) |
> |---|---|---|---|
> | 0% (Baseline) | --- | 1.74 | 0.00 |
> | 20% | SVD-LLM | 2.07 | 0.34 |
> | 20% | ZS-SVD | **1.91** | **0.17** |
> | 40% | SVD-LLM | 2.57 | 0.84 |
> | 40% | ZS-SVD | **2.44** | **0.70** |
> | 60% | SVD-LLM | 3.98 | 2.25 |
> | 60% | ZS-SVD | **3.81** | **2.07** |
>
> Furthermore, in Table 6 of our paper, we perform an ablation study over different singular value pruning strategies within the same whitened space. The results clearly show that the zero-sum strategy consistently achieves the best performance.
>
> ---
>
> ## Q2: Fairness of HQ and Remapping
>
> A more detailed explanation of HQ and remapping is provided in the paper.
>
> **Remapping.** We refer the reviewer to the detailed description in the DOBI-SVD paper.
>
> **HQ.** For pruning ratios above 50%, HQ retains 2x more parameters before truncation and then quantizes them to 8-bit, so the target compression is achieved through truncation plus quantization. Since full 8-bit quantization alone already gives 50% compression, HQ is only applicable if the pruning rate is beyond 50%.
>
> Both remapping and HQ combine SVD truncation with quantization. Finding a better combination of these two approaches remains relatively underexplored in the compression literature, which we consider an interesting direction for future work.
>
> We provide the following analysis at a 60% pruning ratio on LLaMA-7B, comparing ZS-SVD, SVD-LLM, and DOBI-SVD under three schemes: truncation only, vanilla remapping from DOBI-SVD, and HQ.
>
> | Method | Truncation | Remapping | HQ |
> |---|---|---|---|
> | SVD-LLM | 53.74 | 14.98 | 7.95 |
> | DOBI-SVD | 46.18 | 9.95 | 8.79 |
> | ZS-SVD | **45.17** | **9.25** | **6.73** |
>
> HQ consistently outperforms remapping across all methods, and ZS-SVD performs best under all three schemes.
>
> ---
>
> ## Q3: Loss Fluctuation Under Re-Truncation
>
> Validation loss per stage, LLaMA-7B at 60% pruning (WikiText-2). T = Truncation, C = Correction.
>
> | | Uncomp. | T1 | C1 | T2 | C2 | T3 | C3 | T4 | C4 | T5 | C5 | T6 |
> |---|---|---|---|---|---|---|---|---|---|---|---|---|
> | Val. Loss | 1.74 | 3.81 | 3.32 | 3.29 | 3.18 | 3.20 | 3.12 | 3.16 | 3.12 | 3.16 | 3.06 | 3.02 |
>
> Across alternating truncation and correction steps, validation loss shows a decaying zigzag pattern, with each retruncation causing only a small increase and later stages converging to lower loss. This supports the claim that correction updates remain close to the low-rank manifold, so retruncation does not significantly undo the gain.
>
> ---
>
> We hope these clarifications address the reviewer's concerns and would be grateful if this is reflected in the final score.

---

> > ### Author Rebuttal · Reviewer_ZYmV · 2026-04-02
> >
> > Thank you for the detailed rebuttal; my initial concerns have largely been addressed. However, while the proposed method demonstrates empirical effectiveness, the paper as a whole still leans towards an empirical study. For ICML, I believe a complete theoretical proof is highly important. Therefore, I will maintain my original score of 4(Weak Accept).

---

> > > ### Author Response · Authors · 2026-04-03
> > >
> > > We thank the reviewer for the thoughtful feedback and for engaging with our rebuttal. We are glad that the main concerns have been addressed and appreciate the careful evaluation.

---

### Decision · Program_Chairs · 2026-04-30

**Decision:**

Accept (regular)

**Comment:**

This paper introduced a post-training low-rank compression technique, called Zero-Sum SVD (ZS-SVD).  The reviewers agreed that the proposed method is novel and clearly explained, the experimental evaluations are comprehensive to demonstrate its effectiveness. The authors’s rebuttal has successfully addressed concerns on the validity of theorem 3.1, technical limitations, and experimental gaps. Therefore, the paper can be accepted to the conference. The authors have to include the additional experiments and statistic analysis promised in their rebuttal into the final version.